# 1.5 years of TROPOMI CO measurements: Comparisons to MOPITT and ATom

Sara Martínez-Alonso[1], Merritt Deeter[1], Helen Worden[1], Tobias Borsdorff[2], Ilse Aben[2], Róisin Commane[3], Bruce Daube[7], Gene Francis[1], Maya George[4], Jochen Landgraf[2], Debbie Mao[1], Kathryn McKain[5,6], and Steven Wofsy[7]

[1]Atmospheric Chemistry Observations and Modeling (ACOM), National Center for Atmospheric Research(NCAR), Boulder, CO, USA
[2]SRON Netherlands Institute for Space Research, Utrecht, Netherlands
[3]Lamont-Doherty Earth Observatory, Columbia University, NY, USA
[4]LATMOS/IPSL, Sorbonne University, UVSQ, CNRS, Paris, France
[5]Cooperative Institute for Research in Environmental Sciences (CIRES), University of Colorado, Boulder, CO, USA
[6]Earth System Research Laboratory, Global Monitoring Division (GMD), National Oceanic and Atmospheric Administration, Boulder, CO, USA
[7]School of Engineering and Applied Science and Department of Earth and Planetary Sciences, Harvard University, Cambridge, MA, USA

**Correspondence:** Sara Martínez-Alonso (sma@ucar.edu)

**Abstract.** We have analyzed TROPOspheric Monitoring Instrument (TROPOMI) carbon monoxide (CO) data acquired between November 2017 and March 2019 with respect to other satellite (MOPITT, Measurement Of Pollution In The Troposphere) and airborne (ATom, Atmospheric Tomography mission) datasets to understand better TROPOMI's contribution to the global tropospheric CO record (2000 to present). MOPITT and TROPOMI are two of only a few satellite instruments to ever

derive CO from solar reflected radiances. Therefore, it is particularly important to understand how these two datasets compare. Our results indicate that TROPOMI CO retrievals over land show excellent agreement with respect to MOPITT: relative biases and their standard deviation (i.e., accuracy and precision) are on average -3.73 $\pm$ 11.51, -2.24 $\pm$ 12.38, and -3.22 $\pm$ 11.13 %, compared to the MOPITT TIR (thermal infrared), NIR (near infrared), and TIR+NIR (multispectral) products, respectively. TROPOMI and MOPITT data also show good agreement in terms of temporal and spatial patterns.

Despite depending on solar reflected radiances for its measurements, TROPOMI can also retrieve CO over bodies of water if clouds are present, by approximating partial columns under cloud tops using scaled, model-based reference CO profiles. We quantify the bias of TROPOMI total column retrievals over bodies of water with respect to colocated *in situ* ATom CO profiles after smoothing the latter with the TROPOMI column averaging kernels (AK), which account for signal attenuation under clouds (relative bias and its standard deviation = 3.25 $\pm$ 11.46 %). In addition, we quantify $e_{null}$ (the null-space error), which

accounts for differences between the shape of the TROPOMI reference profile and that of the ATom true profile ($e_{null}$ = 2.16 $\pm$ 2.23 %). For comparisons of TROPOMI and MOPITT retrievals over open water we compare TROPOMI total CO columns to their colocated MOPITT TIR counterparts. Relative bias and its standard deviation are 2.98 $\pm$ 15.71 % on average.

We investigate the impact of discrepancies between the *a priori* and reference CO profiles (used by MOPITT and TROPOMI, respectively) on CO retrieval biases by applying a null-space adjustment (based on the MOPITT *a priori*) to the TROPOMI total column values. The effect of this adjustment on MOPITT/TROPOMI biases is minor, typically 1-2 percentage points.

## 1 Introduction

Even though carbon monoxide (CO) constitutes less than one millionth of the troposphere in volume, it is of great importance to understand climate and to monitor and predict air quality. Tropospheric CO is produced by incomplete fuel combustion, biomass burning, and oxidation of methane and other hydrocarbons. CO's main sink is oxidation by the hydroxyl radical (OH) (Spivakovsky et al., 2000; Lelieveld et al., 2016); this reaction produces greenhouse gases such as carbon dioxide and tropospheric ozone. Additionally, OH engaged in reactions with CO is not available to scavenge other greenhouse gases such as methane, which then have a longer lifetime in the atmosphere. As a consequence, CO emissions have a positive indirect radiative forcing of 0.23 W/m$^2$ (Myhre et al., 2013). The mean lifetime of tropospheric CO (variable by season and latitude, in addition to other factors; Holloway et al., 2000) is approximately 2 months. Because of its average lifetime, -long enough to last through horizontal and vertical transport and, yet, short enough not to become well mixed-, it is often used as a tracer to monitor the distribution, transport, sources, and sinks of polluted plumes (e.g., Heald et al., 2003). A self-consistent, uninterrupted record of global tropospheric CO is, thus, key to both climate and air quality studies. The aim of this work is to facilitate the extension of the current satellite record with newly available TROPOMI (TROPOspheric Monitoring Instrument) measurements by evaluating those with respect to satellite MOPITT (Measurements Of Pollution In The Troposphere) and *in situ* ATom (Atmospheric Tomography mission) CO data.

The pre-launch targets for TROPOMI total CO column accuracy and precision were 15 and 10 %, respectively, for both clear and low-altitude-cloud observations (Veefkind et al., 2012; Landgraf et al., 2016). Retrieval errors are expected to be larger for cloudy conditions due to several effects, including the shape of model-based reference profiles (Borsdorff et al., 2018b). Global comparisons of TROPOMI retrievals with respect to ECMWF/IFS (European Center for Medium-Range Weather Forecast/Integrated Forecasting System) CO assimilation results (which incorporate CO retrievals from MOPITT as well as from IASI, the Infrared Atmospheric Sounding Interferometer (Clerbaux et al., 2009)) showed a relative high bias of 3.2 % with standard deviation of 5.5 % (Borsdorff et al., 2018a). TROPOMI CO retrievals over land have also been previously compared to ground-based measurements from nine TCCON (Total Carbon Column Observing Network; Wunch et al., 2011) stations for selected dates between 9 November 2017 and 4 January 2018; good agreement between both datasets was found, with the TROPOMI CO product well within the mission requirements (Borsdorff et al., 2018b). Here we analyze daily global TROPOMI retrievals acquired between 7 November 2017 and 10 March 2019 with respect to MOPITT and ATom.

MOPITT is the only currently operating satellite instrument deriving CO from near-infrared (NIR), thermal-infrared (TIR), and multispectral (TIR+NIR) radiances; also, it has the longest global CO record to date (2000-present). TROPOMI was, until recently, the only other operative satellite instrument retrieving CO from NIR measurements. (ENVISAT SCIAMACHY (2002-2012; Bovensmann et al., 1999) and GOSAT-2 TANSO-FTS-2 (since 2019; NIES, 2019) are two other instances.) Thus, understanding how MOPITT and TROPOMI retrievals compare to each other is important. MOPITT results are systematically validated using airborne vertical profiles (Deeter et al., 2019, and references therein) and ground measurements (Buchholz

et al., 2017; Hedelius et al., 2019), as well as compared to other satellite datasets (Worden et al., 2013a; Martínez-Alonso et al., 2014; George et al., 2015). Thus, its continuity and consistency are well understood.

Despite the low reflectivity of open water, TROPOMI CO retrievals over bodies of water are possible if clouds are present. In these cases partial CO columns under the cloud tops are approximated by scaled TROPOMI reference profiles (Borsdorff et al., 2018b). We quantify the error introduced by this approach by comparing TROPOMI CO retrievals over bodies of water to both airborne ATom-4 (fourth ATom campaign) and MOPITT TIR data.

Next we describe the datasets used (Sect. 2), detail how comparisons were performed (Sect. 3), present results from these comparisons (Sect. 4), discuss their significance (Sect. 5), and offer conclusions (Sect. 6). Additional results are available in the Supplement Materials.

## 2 Data

### 2.1 TROPOMI

TROPOMI is a push-broom imaging spectrometer on board ESA's Sentinel-5 Precursor platform, flying in a sun-synchronous orbit at 824 km altitude and 13:30 LST (local standard time) Equator crossing time. Its swath width of 2600 km allows for global daily coverage at very high spatial resolution, with a 7.2 x 7.2 km$^2$ footprint at nadir (Veefkind et al., 2012). (A change in the Copernicus Sentinel-5P operations scenario postdating the work presented here has resulted in a 7.2 x 5.6 km$^2$ footprint at nadir, starting 6 August 2019.) TROPOMI measures radiances in the ultraviolet, visible, and solar reflected infrared. Total CO column values are obtained from measurements of reflected solar infrared radiation in the 2.3 $\mu$m spectral range (Landgraf et al., 2016), corresponding to the first overtone of the CO stretch fundamental. Over land, retrievals are performed in both clear and cloudy conditions. TROPOMI CO retrievals over bodies of water are possible if clouds are present in the field of view (Landgraf et al., 2016); otherwise, because of the low reflectivity of open water to shortwave infrared solar radiation, insufficient radiance would be available for the instrument to measure. TROPOMI retrievals are achieved by estimating the altitude of the cloud top from the difference between measured and modeled methane, as described in Landgraf et al. (2016), and then approximating the partial CO column under the cloud top by the colocated, scaled TROPOMI reference profile partial column.

TROPOMI CO retrievals are based on SICOR (Shortwave Infrared Carbon Monoxide Retrieval) (Vidot et al., 2012). In this physics-based algorithm, the retrieval state vector includes a single scaling factor representing the ratio of the retrieved CO profile to the reference CO profile (Borsdorff et al., 2014). Reference profiles are generated with the global chemical transport model TM5 (Krol et al., 2005); they are variable with respect to location, month, and year. Retrieved total CO column values simply correspond to the vertically-integrated CO profile. Over land, in the absence of clouds, the TROPOMI total CO column averaging kernel (AK; Fig. 1) is near unity over the entire vertical profile (Landgraf et al., 2016). Thus, clear-sky total CO column retrievals are negligibly affected by either the actual vertical distribution of CO or the shape of the CO reference profile. In the presence of clouds, however, over both land and bodies of water, the total CO column retrievals are mainly sensitive to the above-cloud CO partial column. The lack of sensitivity to the below-cloud CO partial column is compensated

by increasing the sensitivity to the above-cloud CO partial column. Clouds thus lead to total column AK values greater than one above the cloud decreasing towards zero below the cloud (Landgraf et al., 2016).

The earliest TROPOMI CO retrievals date from 7 November 2017; therefore, this is the initial date of the period we analyze here. For any given day, we used either OFFL (offline) or RPRO (reprocessed) files, all from Collection 01, and from the most recent processor version available (10001, 10002, 10100, 10200, 10202, 10301, or 10302).

Retrievals were filtered as follows. The two most westward pixels in each granule were removed to avoid artifacts from unresolved calibration issues (Borsdorff et al., 2018a, b); daytime only observations were selected by keeping those with solar zenith angle <80°. Quality flag values (QA) were used to preserve clear-sky and clear-sky-like observations over land (QA = 1, corresponding to optical thickness <0.5 and cloud height <500 m) or observations with mid-level clouds over bodies of water (QA = 0.5; optical thickness ≥ 0.5 and cloud height <5000 m) (Landgraf et al., 2018).

## 2.2 MOPITT

MOPITT is a cross-track scanning gas correlation radiometer on board NASA's Terra satellite (Drummond and Mand, 1996; Drummond et al., 2010; Worden et al., 2013b). Terra is in a sun-synchronous orbit at 705 km altitude and 10:30 LST Equator crossing time. MOPITT has horizontal resolution near 22 x 22 km$^2$ at nadir and a swath width of 640 km; global coverage is achieved in approximately 3 days. MOPITT observations enable retrievals of tropospheric CO vertical profiles and corresponding total column amounts from both TIR and NIR measurements in the spectral regions where the fundamental (~4.7 $\mu$m) and first overtone (~2.3 $\mu$m) of the CO stretch occur, respectively. TIR measurements are useful over both bodies of water and land, day and night; NIR radiances only in daytime observations over land. MOPITT CO retrieval products are available in three variants (TIR-only; NIR-only; and TIR+NIR, or multispectral) characterized by different vertical sensitivity and random retrieval noise (Deeter et al., 2019, and references therein).

Unlike TROPOMI's, the MOPITT retrieval algorithm relies on optimal estimation whereby *a priori* information constrains the retrieved profile in the absence of information from the measured radiances (Deeter et al., 2003). MOPITT *a priori* profiles vary seasonally and geographically according to a multi-year (2000-2009) Community Atmosphere Model with Chemistry (CAM-Chem) model-based CO climatology (Lamarque et al., 2012). MOPITT profile retrievals are performed on a ten-level pressure grid; the reported retrieval for each level indicates the mean volume mixing ratio (VMR) in the layer immediately above that level. Reported total CO column values are obtained by integrating the retrieved VMR profiles from the surface to the top of the atmosphere. Internally, CO concentrations in the retrieval state vector are represented in terms of the logarithm of the VMR. For each retrieved CO profile, both the full retrieval AK matrix and total column AK are produced simultaneously and are provided as diagnostics. As indicated by the AK (Fig. 1), sensitivity characteristics of the three products are quite different (Deeter et al., 2012). With respect to vertical sensitivity, the total column AK for the NIR-only product are most similar in shape to the TROPOMI total column AK, but NIR retrievals can be significantly constrained by the *a priori*. In comparison, TIR-only total column AK exhibit weaker sensitivity to CO near the surface, but TIR retrievals are less strongly weighted by the *a priori* overall. TIR+NIR total column AK are typically more uniform than for TIR-only retrievals, although the benefits of combining TIR and NIR measurements are only apparent in daytime observations over land.

Here we use daytime archive MOPITT data from version 8 (Deeter et al., 2019); among other improvements, V8 products do not exhibit a latitudinal dependence in partial CO column biases observed in V7. The MOPITT retrieval algorithm processes only clear-sky observations (Francis et al., 2017). The clear/cloudy status of an observation is typically determined from MO-PITT radiances as well as a cloud mask (Ackerman et al., 1998) based on simultaneous observations by MODIS (MODerate resolution Imaging Spectroradiometer, also on board the Terra platform). The ~480 MODIS observations at 1 x 1 km$^2$ horizontal resolution acquired at the same time as a single MOPITT observation and within the MOPITT footprint are identified and collected; relevant MODIS cloud descriptors (available in the MOPITT L2 product) are evaluated. MOPITT observations for which at least 95% of the colocated MODIS cloud mask values are considered clear are passed to the retrieval algorithm. MOPITT archive data are those corrected with gain and offset values derived from an interpolation performed between two consecutive hot-calibration events, which are usually executed once per year. This retrospective correction alleviates large differences in total column values otherwise observed in NIR retrievals; TIR products are affected to a much lesser degree (Deeter et al., 2017). Here we use MOPITT archive data produced after the hot calibration performed between 11 and 23 March 2019; thus, the closing date for the period analyzed here is 10 March 2019. Total column validation results for version 8 products indicate that relative biases and standard deviations are less than 1 and 7 %, respectively (i.e., less than 0.5 and 1.5 x 10$^{17}$ molec. cm$^{-2}$) (Deeter et al., 2019).

## 2.3 ATom-4

To analyze TROPOMI retrievals over bodies of water we use ATom (Wofsy et al., 2018) *in situ* CO profiles from its fourth campaign, carried out between 24 April and 21 May 2018. During ATom-4 more than 150 vertical profiles were acquired, most of them over water in the Atlantic and Pacific regions, and covering a wide latitudinal range. CO concentrations along those profiles were measured with the Harvard QCLS (pulsed-Quantum Cascade Laser System) instrument (Santoni et al., 2014; McManus et al., 2010) and the NOAA Picarro Cavity Ring Down Spectrometer (Crosson, 2008; Karion et al., 2013), both on board NASA's DC-8 platform. Measurements were acquired from 0.2 to 12 km altitude at 1 Hz sampling rate. The QCLS instrument operates in the 4.59 $\mu$m region, with precision and accuracy of 0.15 and 3.5 ppb, respectively (Santoni et al., 2014). The NOAA Picarro measures radiation in the 1.57 $\mu$m region, where the second overtone of the CO stretch is located; the estimated total uncertainty of its measurements is 5.0 ppb at 1 Hz, or 3.4 ppb at 0.1 Hz (McKain and Sweeney, 2018). Here we use the merged QCLS-Picarro data product CO.X from the dataset version published 28 March 2018 and updated 25 November 2019. The quantity CO.X uses QCLS CO data with calibration gaps filled in by Picarro CO data, after subtracting the low-pass filtered difference between the QCLS and the somewhat noisier Picarro measurement. Both instruments were calibrated to the NOAA X2014A CO scale. Measurements account for drift of CO in their field calibration tanks (ESRL, 2018) by having them measured at the central calibration laboratory before and after the campaign and applying a linear drift correction to the assigned values.

## 3 Methods

In Sect. 4 we separately present quantitative comparisons of TROPOMI total column retrievals with MOPITT total column retrievals and with *in situ* profiles measured from aircraft. However, different methods are required in each case (Rodgers and Connor, 2003). Comparisons with *in situ* profile data are generally simpler and more easily interpreted, because the vertical sensitivity of the satellite measurement can be represented exactly using the retrieval AK.

### 3.1 MOPITT and TROPOMI algorithm differences

Fundamental differences in the MOPITT and TROPOMI retrieval algorithms result in a challenge to find consistent intercomparison methods. The MOPITT algorithm is based on optimal estimation as developed by Rodgers (2000). TROPOMI uses a profile-scaling algorithm based on Tikhonov regularization, as described in Vidot et al. (2012), Borsdorff et al. (2014), Landgraf et al. (2016), and references therein. Moreover, the MOPITT state vector and AK are based on CO profiles of log(VMR) whereas the TROPOMI retrieval algorithm involves CO profiles expressed in column density values (molecules per unit area). For simplicity, we assume in the following discussion that MOPITT log(VMR)-based quantities can be converted to column density-based quantities.

Thus, neglecting error terms, we can write for MOPITT

$$c^{MOP} \approx a^{MOP} x_{true} + (C - a^{MOP}) x_a^{MOP} \tag{1}$$

where $c^{MOP}$ is the retrieved total column, $a^{MOP}$ is the column density-based total column AK, $x_{true}$ is the true profile, $C$ is the total column operator, and $x_a^{MOP}$ is the *a priori* profile. $c^{MOP}$, $x_{true}$, and $x_a^{MOP}$ are all expressed in column density (molecules per unit area). $C$ and $a^{MOP}$ are dimensionless.

For TROPOMI, however, we have

$$c^{TROP} \approx a^{TROP} x_{true} \tag{2}$$

where $c^{TROP}$ is the retrieved total column and $a^{TROP}$ is the total column AK. Thus, the retrieved total column for MOPITT partially depends on a "null-space contribution" given by the term *(C - $a^{MOP}$) $x_a^{MOP}$* whereas the TROPOMI total column retrieval lacks this term. For MOPITT, this term represents the weighting of the MOPITT *a priori* profile in the retrieved total column. As noted in Borsdorff et al. (2014), a null-space contribution term is not beneficial for data assimilation applications, but may be added to the TROPOMI total column retrieval by the user if a particular source of *a priori* information is desired. This option is applied in Sect. 4.1.4 and 4.2.3 as a means of testing the influence of the *a priori* profile on MOPITT/TROPOMI comparisons.

### 3.2 *In-situ* validation: TROPOMI versus ATom-4

*In-situ* profile data acquired from aircraft are well-suited for validating satellite CO retrievals. In the following we use the ATom-4 *in-situ* dataset, which mainly includes over-ocean observations. We derived both true and retrieval-simulated (i.e., un-

smoothed and smoothed) total CO column values from the ATom-4 profiles; smoothed values account for the vertical sensitivity of the TROPOMI measurements as expressed by their AK.

Prior to obtaining unsmoothed/smoothed ATom-4 total CO columns, complete (e.g., from the surface to the top of the atmosphere) ATom-4 CO profiles were generated following the standard method for MOPITT validation with airborne data. Profiles that did not cover the 400-to-800 hPa range were rejected. The remaining profiles (between $271 \pm 48$ hPa and $983 \pm 32$ hPa) were interpolated to match the MOPITT *a priori* 35-level vertical grid, which preserves high vertical resolution in the troposphere. Empty levels at the bottom of each interpolated profile (levels with no CO value) were filled with the interpolated measurement closest to the surface. Similarly, empty levels between the top of the interpolated profile and the tropopause were filled with the interpolated measurement closest to the tropopause. Finally, empty levels above the tropopause were filled with colocated MOPITT *a priori* CO values. Unsmoothed ATom-4 total CO column values were then calculated as follows:

$$x_{true} = 2.12 \times 10^{13} \Delta p VMR \tag{3}$$

where $x_{true}$ is expressed as an array of partial column values in molec. cm$^{-2}$, the constant *2.12 × 10$^{13}$* is in molec. cm$^{-2}$ hPa$^{-1}$ ppbv$^{-1}$, $\Delta p$ is the array of partial column pressure thicknesses in hPa, and *VMR* is the array of VMR values in ppbv units. The derivation of Eq. 3 can be found in Deeter (2009).

Smoothed ATom-4 total CO column values involve the TROPOMI AK. TROPOMI total column retrievals in cloudy scenes are more sensitive to CO above the cloud than to CO below the cloud; smoothed total column values account for this effect explicitly. Similarly to Eq. 2, smoothed ATom-4 CO profiles can be calculated substituting $x_{true}$ by the the complete ATom-4 profiles obtained as detailed above and interpolated to match the 50-level vertical grid of their colocated TROPOMI total column AK. Finally, smoothed ATom-4 total CO column values are calculated applying Eq. (3).

Comparisons between TROPOMI total column retrievals and true (unsmoothed) ATom-4 total column values are the most direct, but they are subject to various sources of random and systematic error. Comparisons between TROPOMI total column retrievals and retrieval-simulated (smoothed) ATom-4 column values should be less affected by TROPOMI vertical sensitivity variations, and can be used to investigate the overall performance of the retrieval. Relative bias values were calculated with respect to ATom in all cases (100x(TROPOMI-ATom)/ATom); column bias values too (TROPOMI-ATom).

In addition, we quantified the error introduced by approximating the partial column below cloud top with the TROPOMI reference profile by calculating the null-space error of the TROPOMI retrieval process ($e_{null}$) as described in Borsdorff et al. (2014) and Landgraf et al. (2016):

$$e_{null} = (C - a^{TROP})x_{true} \tag{4}$$

As discussed in Sect. 4.2.1, analysis of $e_{null}$ may be useful for diagnosing retrieval errors over cloudy scenes related to the shape of the TROPOMI model-calculated reference profiles.

### 3.3 Satellite comparisons: TROPOMI versus MOPITT

#### 3.3.1 Sources of error

Satellite-based retrievals of CO total column, like other remote sensing retrievals, are subject to several sources of error (Rodgers, 2000). Prominent sources of error for both MOPITT and TROPOMI include smoothing error (related to the departure of the total column AK from the ideal dependence, which would have a value of 1 at all altitudes) and random retrieval noise. Other potentially important effects which are not considered further include model parameter error and forward model error (Rodgers, 2000). Retrieval averaging can be used to reduce the effects of retrieval noise but does not reduce smoothing error. Smoothing error is instrument-dependent; it also depends on details of the retrieval algorithm. For both MOPITT and TROPOMI, the total column smoothing error is related to the total column AK and true CO profile, similarly to what Eq. 4 shows.

As discussed in Sect. 2.1, smoothing error for TROPOMI retrievals in clear-sky scenes over land is generally very small since $a^{TROP}$ is near 1 at all altitudes. In scenes containing clouds, which includes all TROPOMI retrievals over the ocean, $a^{TROP}$ increases to values greater than 1 above the cloud and decreases to less than 1 below the cloud (Fig. 1). However, as a result of the profile-scaling method used by TROPOMI, smoothing error also vanishes if the shape of the true profile converges with the shape of the assumed reference profile, even in cloudy scenes (Borsdorff et al., 2014). Smoothing error for TROPOMI will thus be largest in cloudy scenes where the reference profile and true profile exhibit a significant difference in shape.

Smoothing error associated with the MOPITT total column AK, discussed in Sect. 2.2, varies for the TIR-only, NIR-only and TIR+NIR products. However, as indicated by Fig. 1, total column smoothing error for all MOPITT variants will typically be larger than for TROPOMI, because of significant differences of $a^{MOP}$ from the ideal column AK.

Methods for comparing remote sensing retrievals of geophysical quantities (such as trace-gas vertical profiles) from different instruments are described in Rodgers and Connor (2003). Effects that contribute to differences in retrieved values include the use of different *a priori* information for each instrument, differences in AK, and differences in instrument noise. One goal of the described methods is to determine whether or not observed differences in retrievals for two instruments are statistically consistent with known differences in *a priori*, AK, and instrument noise. However, this goal is elusive because technically it also requires knowledge of the statistics (mean and variability) of the ensemble of true atmospheric states being used for the comparisons; this information is often unknown.

Our main goal in performing comparisons of MOPITT and TROPOMI total column retrievals is to quantify differences between the two retrieval products available to users, rather than quantify the actual bias of either product. This goal is addressed by direct "end to end" comparisons of the two untransformed products in various geographical regions, after appropriate matching of the MOPITT and TROPOMI retrievals in space and time. These comparisons quantify the MOPITT/TROPOMI difference statistics (e.g., bias and standard deviation) due to all effects: AK differences, *a priori* differences, and instrument noise.

A secondary goal of the comparisons is to specifically investigate the influence of *a priori* information on MOPITT/TROPOMI retrieval differences. Unlike the AK, which depend fundamentally on characteristics of the instrument, the source of *a priori*

(or reference profiles, in the case of TROPOMI) is a choice of the retrieval algorithm developers. The method for addressing this goal described in Rodgers and Connor (2003) assumes that both retrievals exhibit a similar *a priori* dependence, represented by Eq. 1, and is thus not applicable to TROPOMI. An alternative strategy, suggested in Borsdorff et al. (2014), is to add a null-space contribution $c_{null}^{TROP}$ to the TROPOMI total column retrievals based on the MOPITT *a priori* profile, i.e.,

$$c_{adj}^{TROP} = c^{TROP} + c_{null}^{TROP} = a^{TROP} x_{true} + (C - a^{TROP}) x_a^{MOP} \tag{5}$$

where $c_{adj}^{TROP}$ is the null-space adjusted TROPOMI total column. The adjustment term $c_{null}^{TROP}$ effectively uses the MOPITT *a priori* profile to estimate the CO partial column for profile levels where the TROPOMI measurement lacks sensitivity. This term vanishes when $a^{TROP}$ approaches $C$ and when $x_a^{MOP}$ approaches the TROPOMI reference profile $x_{ref}^{TROP}$ (because $a^{TROP} x_{ref}^{TROP} = C x_{ref}^{TROP}$). For MOPITT/TROPOMI comparisons, this adjustment to the TROPOMI retrieved total columns should reduce differences due to discrepancies between the MOPITT *a priori* profile and TROPOMI reference profile, but should have no effect on differences related to discrepancies in retrieval AK or other sources of retrieval bias. Results of MOPITT/TROPOMI comparisons incorporating this adjustment over land and oceanic regions are presented in Sect. 4.1.4 and 4.2.3, respectively.

### 3.3.2 Land retrieval comparisons

Over land, MOPITT and TROPOMI total column retrievals were compared in clear-sky scenes only. In such scenes, TROPOMI smoothing error is typically negligible since $a^{TROP}$ is close to 1 at all altitudes. For these comparisons, we selected six ROIs (regions of interest; Fig. 2) representative of either polluted or clean regimes. Polluted ROIs include: south-eastern USA (thereafter referred to as USA; 35°N, 95°W to 40°N, 75°W), central Europe (Europe; 45°N, 0°E to 55°N, 15°E), northern half of the Indian Subcontinent (India; 20°N, 70°E to 30°N, 95°E), and north-eastern China (China; 30°N, 110°E to 40°N, 123°E). Clean ROIs are: northern Africa and Arabia (Sahara; 15°N, 20°W to 30°N, 50°E) and western Australia (Australia; 32°S, 112°E to 17°S, 138°E). Two additional ROIs were defined to represent most of the northern and southern (N and S) hemispheres (0°N to 60°N and 60°S to 0°N, respectively). TROPOMI and MOPITT retrievals covering each of these ROIs for the period between 7 November 2017 and 10 March 2019 were gathered and filtered to keep only clear daytime data over land.

Colocated and non-colocated retrievals from the two instruments were analyzed separately; results from the former are presented in Sect. 4.1, whereas supporting results from the latter are presented in the Supplement Materials. We apply the term 'colocated' to pairs of retrievals from two different datasets acquired on the same day and within $\leq 50$ km in horizontal distance. In contrast, we apply the term 'non-colocated' to retrievals from two different datasets acquired on the same day and inside the same ROI. Colocated samples allow for a more direct comparison, since they are more closely representative of the same atmospheric conditions. By using non-colocated retrievals we maximized the size and diversity of the populations analyzed.

Daily scatterplots for each ROI were obtained from the colocated retrievals. We quantified, among others, daily bias (i.e., accuracy) and standard deviation (i.e., precision; calculated from individual biases between each pair of colocated observations) between TROPOMI and each of the three MOPITT products (TIR, NIR, and TIR+NIR). Relative bias values (in %) were

calculated with respect to MOPITT in all cases (100×(TROPOMI-MOPITT)/MOPITT). Column bias values (in molec. cm$^{-2}$),
also provided for completeness, were calculated with respect to MOPITT (TROPOMI-MOPITT). Thus, a negative bias would
indicate that TROPOMI CO values are lower than their MOPITT counterparts.

Results from an analogous comparison of colocated MOPITT and null-space adjusted (as described in Sect. 3.3.1) TROPOMI
total column retrievals can also be found in Sect. 4.1.4.

### 3.3.3 Water retrieval comparisons

Two types of MOPITT/TROPOMI comparisons were made over oceanic regions. Direct comparisons, performed without any
adjustments to either the MOPITT or TROPOMI total column values, are presented in Sect. 4.2.2. Comparisons incorporating
the TROPOMI null-space adjustment, as described in Sect. 3.3.1, are presented in Section 4.2.3. Statistics for the Northern
and Southern Hemispheres are analyzed separately. Given their nature, all comparisons over bodies of water used colocated
observations.

## 4   Results

Land-only comparisons have the purpose of evaluating TROPOMI's performance with respect to MOPITT TIR, NIR, and
TIR+NIR. Separate comparisons were performed using either colocated data (results in Sect. 4.1; for untransformed and null-
space adjusted TROPOMI) or non-colocated data (Supplement Materials). Water-only comparisons aim to estimate the error
introduced in TROPOMI retrievals over bodies of water, only possible in cloudy conditions, by approximating CO concen-
295 trations below cloud top by colocated, scaled TROPOMI reference profile values. Two sets of water-only comparisons were
performed. First, with respect to *in situ* ATom-4 profiles, accounting for differences in TROPOMI vertical sensitivity as rep-
resented by its AK (Sect. 4.2.1). Second, we compared untransformed TROPOMI with respect to MOPITT TIR total column
values (Sect. 4.2.2). Third, we compared null-space adjusted TROPOMI with respect to MOPITT TIR total column values
(Sect. 4.2.3). Additional comparisons with respect to MOPITT TIR and ATom-4 profiles assuming a simple scenario where
TROPOMI only had sensitivity to CO above cloud top are available in the Supplement Materials; this approximation would be
most accurate for optically thick clouds.

### 4.1   TROPOMI retrievals over land

Here we describe results from the comparison of daily (from 7 November 2017 to 10 March 2019) colocated TROPOMI and
MOPITT retrievals over 8 ROIs: 2 hemispheric, 4 representative of polluted regions, and 2 of clean regions (Fig. 2). Daily bias
and standard deviation values calculated between TROPOMI and each of the three MOPITT products are presented below.

### 4.1.1   TROPOMI versus MOPITT TIR

Daily results from the analysis of colocated TROPOMI and MOPITT TIR data (Fig. 3) show that during the ~1.5 years
analyzed, TROPOMI and MOPITT TIR total CO column retrievals were close to each other both in magnitude and temporal

variation. Both datasets agree in displaying strong differences between clean ROIs (Sahara and Australia; 10-20 x $10^{17}$ molec. cm$^{-2}$) and highly polluted ROIs (India and China; 15-40 x $10^{17}$ molec. cm$^{-2}$). They also show the expected differences between the two hemispheres: retrievals are, overall, lower in the S Hemisphere ROI (10-20 x $10^{17}$ molec. cm$^{-2}$ versus 15-22 x $10^{17}$ molec. cm$^{-2}$) due to less land area, population, and industrial activity. Both TROPOMI and MOPITT TIR show similar seasonal variability. ROIs located in the northern hemisphere present an absolute maximum during boreal winter and a secondary maximum in late boreal summer. The absolute maximum is consistent with winter CO accumulation due to shorter days and (at high latitudes) larger solar zenithal angles resulting in less photolysis, and to increased emissions due to biomass burning north of the Equator in Africa. The secondary maximum is most likely due to fire emissions. Conversely, seasonal trends in southern hemisphere ROIs show a maximum in September-October, consistent with CO accumulation during austral winter and emissions from biomass burning S of the equator.

Daily relative bias values are generally within a $\pm10$ % range for all the ROIs except the two most polluted, India and China (Fig 3.e and 3.f), where biases reach higher values, mostly in the -20 to 20 % range. When averaged over time (Table 1), relative biases are between -8.15 % (Sahara) and 3.55 % (China), with a mean for all the ROIs of -3.73 %. We note that biases for most ROIs are predominantly negative, except for China, where most daily biases are positive. Averaged relative standard deviation values per ROI are between 6.05 and 16.04 % (USA and S Hemisphere, respectively), with a mean for all ROIs of 11.51 %.

### 4.1.2 TROPOMI versus MOPITT NIR

Figure 4 shows daily results from the comparison of colocated TROPOMI and MOPITT NIR land retrievals; time-averaged results are summarized in Table 1. The ranges of daily mean retrievals and seasonal trends observed in each ROI are in general analogous to those described in Sect. 4.1.1. Relative bias values averaged for the period analyzed range between -7.93 % (USA) and 2.86 % (Sahara), while the mean for all the ROIs is -2.24 %. Daily relative bias values for the Sahara ROI (-5 to 12 % range; Fig. 4.g) differ strongly from those calculated with respect to MOPITT TIR (Fig. 3.g) (-12 to -5 % range). For all the other ROIs, relative biases with respect to MOPITT NIR are broadly similar in magnitude to those with respect to MOPITT TIR, albeit the former present larger oscillations with time. This is consistent with the MOPITT NIR retrievals being more sensitive to geophysical noise due to changes in albedo during a MOPITT observation associated with spacecraft motion (Deeter et al., 2011). Relative standard deviation values averaged over time are between 9.95 and 16.15 % (USA and China, respectively), with a mean for all ROIs of 12.38 %.

### 4.1.3 TROPOMI versus MOPITT TIR+NIR

Daily results from colocated TROPOMI and MOPITT TIR+NIR retrievals are shown in Fig. 5; time-averaged results are summarized in Table 1. Results are similar to those described in Sect. 4.1.1 in terms of daily mean retrieval values, retrieval seasonal trends, and relative biases. The latter range between -7.94 % (Sahara) and 4.53 % (China); the mean for all ROIs is -3.22 %. Averaged relative standard deviation values are between 6.48 % (Sahara) and 15.68 % (S Hemisphere), with a mean for all ROIs of 11.13 %.

#### 4.1.4 Null-space adjusted TROPOMI versus MOPITT

Table 4 summarizes time-averaged bias values resulting from the comparison of colocated, null-space adjusted TROPOMI and MOPITT land retrievals. Relative bias values averaged for all ROIs are -2.52, -1.07, and -1.99 % (for MOPITT TIR, NIR, and TIR+NIR, respectively). Similarly, averaged relative standard deviation values are 11.57, 12.40, and 11.21 %. Daily results are
analogous to those shown in Fig. 3, 4, and 5 both in magnitude and temporal variation.

### 4.2 TROPOMI retrievals over water

Next we present results from the comparison of colocated TROPOMI and ATom-4 retrievals between 24 April and 21 May 2018 over the Atlantic and Pacific regions. Similarly, we describe results obtained from colocated TROPOMI (both untransformed and null-space adjusted) and MOPITT TIR over-water retrievals acquired between 7 November 2017 and 10 March 2019 over
the two hemispheric ROIs. The ATom-4 data offer the opportunity to compare TROPOMI retrievals to *in situ* measurements; the MOPITT dataset has the advantage of a substantially larger number of samples, distributed over a longer period of time and a wider geographical area.

#### 4.2.1 TROPOMI versus ATom-4

Results from the TROPOMI and ATom-4 comparison over bodies of water are summarized in Fig. 6 and Table 2. As described
in Sect. 3.2, comparisons were performed both in terms of true (unsmoothed) and retrieval-simulated (smoothed) ATom-4 total column values; the latter account for the vertical sensitivity of the TROPOMI retrievals. Figure 6.a shows that unsmoothed ATom-4 total CO columns and TROPOMI are strongly correlated (R = 0.93, slope of linear fit = 0.96) and exhibit a negative relative bias (-4.76 %) indicative of low TROPOMI values with respect to the true ATom-4. In contrast, Fig. 6.b shows results for smoothed ATom-4 versus TROPOMI. The relative bias is in this case better (3.25 %) and the fit between the two datasets has a slightly larger R (0.94), indicative of an improved correlation. The slope of the linear fit is, however, slightly lower
(0.90). Figure 7 shows the smoothed ATom-4 values in the context of TROPOMI; TROPOMI clearly captures the geographical patterns of the *in situ* measurements. Relative biases show no latitudinal dependence (Fig. 8).

As seen in Sect. 3.2, we can separately quantify the expected difference between the true total column and the TROPOMI retrieved total column due to the differences in shape between the true profile and the TROPOMI reference profile. In clear-sky
scenes (over land), the TROPOMI radiances fundamentally measure the integrated total column and the shape of the reference profile does not significantly affect the accuracy of the retrieved total column. In cloudy scenes (over land or water), however, the total column retrieval becomes more sensitive to above-cloud CO than to below-cloud CO; the validity of the reference profile shape acts in this case as a source of retrieval error. Values of the null-space error ($e_{null}$) calculated for each ATom-4 profile using Eq. 4 versus latitude are shown in Fig. 9. The relative mean and standard deviation values of $e_{null}$ calculated with
respect to true (unsmoothed) ATom-4 total columns are $2.16 \pm 2.23$ % (i.e., $3.70 \pm 3.75 \times 10^{16}$ molec. cm$^{-2}$). The prevalence of positive values for $e_{null}$ indicates that, on average, the reference profiles analyzed have a slight tendency to have too much CO

near the surface, resulting in an overestimate of the below-cloud partial column. No clear latitudinal dependence is observed in $e_{null}$.

### 4.2.2 TROPOMI versus MOPITT TIR

Figure 10.a and Table 3 summarize results from our comparison of colocated TROPOMI and MOPITT TIR retrievals over bodies of water in the N Hemisphere ROI. Relative biases are small (3.82 % on average); the standard deviation of the biases is 13.27 % on average. Results for the S Hemisphere ROI are summarized in Fig. 10.b and Table 3. Relative biases and their standard deviation values are similarly small (2.14 % and 18.15 % on average). As expected, retrievals are higher in the N Hemisphere, due to larger emissions from the continents in that hemisphere. Seasonal patterns in daily CO means are

analogous to those described for the two hemispheric land ROIs.

### 4.2.3 Null-space adjusted TROPOMI versus MOPITT TIR

Table 5 summarizes time-averaged bias values resulting from the comparison of colocated, null-space adjusted TROPOMI and MOPITT TIR retrievals over bodies of water. Relative bias values averaged for the period analyzed are 5.90 and 3.82 % (N and S Hemisphere ROIs, respectively); averaged relative standard deviation values are 13.19 and 18.11 %. Daily results are

analogous in magnitude and temporal variation to those shown in Fig. 10.

## 5 Discussion

TROPOMI and MOPITT are consistent with each other in terms of the main spatial and seasonal CO features they capture, as shown by mean seasonal maps (Fig. 11). Both datasets display relatively high values in the Northern hemisphere during boreal winter (panels .a and .b) and spring (.c and .d), similarly high values during all seasons in Africa and Asia, and relatively

high values due to Amazon fires in austral summer and fall (.a and .b, .g and .h). We note differences between TROPOMI and MOPITT that we interpret as due to their contrasting daytime passing times (1:30PM and 10:30AM, respectively): TROPOMI shows higher CO over Africa than MOPITT, consistent with higher CO emissions from afternoon fires than from morning fires. (Fires are commonly more active in the afternoon than in the morning, as observed in fire counts from same day morning Terra MODIS versus afternoon Aqua MODIS (Giglio et al., 2006).) We also note that TROPOMI retrievals over Amazonia are

lower than MOPITT's in all seasons. Identifying the reason for this discrepancy will require further investigation.

Quantitative results from the analysis of colocated TROPOMI and MOPITT land retrievals, summarized in Fig. 12 and Table 1, also show good agreement. Relative biases for all ROIs (-3.73 $\pm$ 11.51, -2.24 $\pm$ 12.38, and -3.22 $\pm$ 11.13 % compared to MOPITT TIR, NIR, and TIR+NIR, respectively) are well within TROPOMI's required 15 % accuracy and close to 10 % precision target (Veefkind et al., 2012; Landgraf et al., 2016). We note that biases are mostly negative (i.e., TROPOMI retrievals

are lower than MOPITT); further analyses would be needed to explain this observation. One exception is China, where biases are predominantly positive. Statistical results obtained from each of the three MOPITT products are consistent with each other for all the ROIs, except for the Sahara. In this case, relative biases between TROPOMI and MOPITT NIR are positive and

closer to zero than biases between TROPOMI and TIR or TIR+NIR products. Results from non-colocated retrievals, available in the Supplement Section and summarized in Fig. 13, reinforce all these observations and provide additional insight.

Several factors may contribute to the contrasting results for the China ROI. First, because of its superior spatial resolution (7.2 x 7.2 km$^2$), TROPOMI can resolve small, highly polluted plumes which would appear diluted at MOPITT's 22 x 22 km$^2$ resolution. Second, TROPOMI provides daily global coverage, while MOPITT's return period is approximately three days; as a result, TROPOMI has more opportunities to sample highly polluted areas than MOPITT. Third, conservative MOPITT cloud mask rules may be responsible for fewer MOPITT retrievals over highly polluted regions, which are frequently hazy due to

aerosols. Detailed daily maps (e.g., Fig. 14) obtained in the analysis of non-colocated observations indicate that MOPITT often fails to retrieve over highly polluted areas like Beijing (China). In this example many MOPITT observations, despite having been classified as cloud-free based on MOPITT radiances, were labeled cloudy (and no retrieval was performed) based on the MODIS cloud mask, which may be interpreting haze due to pollution or fire smoke as clouds. We note that comparisons of non-colocated retrievals are more strongly affected by these factors; this is consistent with particularly high positive biases

derived from non-colocated retrievals over China (Fig. 13).

    Possible causes for the contrasting relative biases obtained from the MOPITT NIR product over the Sahara include aerosol and/or surface albedo effects. Further work is needed to diagnose these effects for different wavelengths and to account for differences between MOPITT and TROPOMI measurement and retrieval methods. Determining the most accurate retrievals would require *in situ* CO column measurements (e.g., airborne profiles) that are not currently available for that region.

There appears to be a seasonal component in MOPITT/TROPOMI bias values in the two hemispheric ROIs and Australia. Polluted ROIs (USA, Europe, India, China) and the Sahara do not seem to be affected (Fig. 3, 4, and 5). Biases between MOPITT and null-space adjusted TROPOMI retrievals show the same seasonal component, indicating that it is not caused by the MOPITT *a priori*. The seasonal variability of MOPITT has been validated in the past using ground-based measurements. In their comparison to NDACC data (Network for the Detection of Atmospheric Composition Change; De Maziere et al., 2018),

Buchholz et al. (2017) found no significant seasonally dependent bias for MOPITT products. Hedelius et al. (2019) compared MOPITT to the TCCON dataset, reporting no persistent seasonal trend globally and some seasonal variability for individual sites. Further work will be needed to identify the origin of a possible seasonal component in MOPITT/TROPOMI bias values.

    We have also analyzed daytime, colocated TROPOMI and ATom-4 data over the Atlantic and Pacific regions for the period between 24 April and 21 May 2018 to quantify the error introduced in TROPOMI retrievals over bodies of water (possible only

under cloudy conditions) by approximating below-cloud-top partial columns with their colocated, scaled reference profiles. There is excellent agreement (-4.76 ± 11.15 % relative bias, i.e., below the mission requirement of 15 % accuracy and close to the 10 % precision target (Veefkind et al., 2012; Landgraf et al., 2016)) between ATom-4 total columns calculated from the true (unsmoothed) *in situ* profiles and the reported TROPOMI total columns (Fig. 6.a). Retrieval-simulated ATom total CO column values are even closer to the TROPOMI retrievals (3.25 ± 11.46 % relative bias); this comparison accounts for the actual

vertical sensitivity of the retrieval process as expressed in the TROPOMI AK, and summarizes the overall performance of the retrievals. The relative contributions of e$_{null}$ with respect to true ATom-4 total CO columns are small (2.16 ± 2.23 %) and

mostly positive, indicating a slight overestimate of the below-cloud partial column in the cases analyzed. No clear latitudinal dependence is observed in relative biases of total CO column or in $e_{null}$.

For an analysis of TROPOMI retrievals over bodies of water representative of a longer period of time (7 November 2017 to
10 March 2019) and larger region (N and S Hemisphere ROIs), we used colocated MOPITT TIR observations. Untransformed TROPOMI retrievals result in relative bias values of 2.98 % on average; relative standard deviation of the bias are 15.71 % on average.

The main goal of the MOPITT/TROPOMI comparisons was to quantify differences using the untransformed retrievals; results have been discussed above. A secondary goal was to analyze the contributions of different sources of retrieval bias.
Two fundamental sources are differences in vertical sensitivity, as defined by the total column AK, and differences between the MOPITT *a priori* and TROPOMI reference profiles. We estimated the error due to differences between the shape of the TROPOMI reference profile and that of the ATom true profile by calculating $e_{null}$ respect to ATom-4 measurements; this error is in the order of 2 %. Without knowing the true CO profiles, there is no obvious way to quantify how differences in the total column AK influence the MOPITT/TROPOMI retrieval differences. We can, however, use the null-space adjustment
technique to examine how sensitive MOPITT/TROPOMI differences are to *a priori*/reference profile discrepancies. Our results indicate that biases between MOPITT and null-space adjusted TROPOMI retrievals (Tables 4 and 5) are very close to biases between MOPITT and TROPOMI untransformed retrievals (Tables 1 and 3). By accounting for differences between *a priori* and reference profiles, the absolute value of relative biases over land decrease by 1.21, 1.17, and 1.23 percentage points, or p.p., on average (for MOPITT TIR, NIR, and TIR+NIR, respectively). The change in relative standard deviation values is also very
small (0.06, 0.02, and 0.08 p.p. on average). Similarly, relative biases over bodies of water change by 1.88 p.p. on average; the change in relative standard deviation values is 0.06 p.p. on average. To sum up, the error introduced by discrepancies between MOPITT *a priori* profiles and TROPOMI reference profiles is very small, near 1-2 p.p. As expected, this error is slightly larger under cloudy conditions, as is the case in TROPOMI retrievals over water.

## 6   Conclusions

A consistent global record of tropospheric CO is important for climate studies as well as for air quality monitoring and prediction. To better understand TROPOMI in the context of the current CO satellite record and thus facilitate the record's extension, we have compared TROPOMI data to other satellite (MOPITT) and airborne (ATom) datasets. Our results show that the accuracy of TROPOMI retrievals with respect to MOPITT and ATom far exceeds Sentinel-5P mission requirements (Veefkind et al., 2012; Landgraf et al., 2016). The precision values calculated for some of the ROIs analyzed surpass the target value by
a few percent.

We have analyzed cloud-free, land-only TROPOMI and MOPITT retrievals from 7 November 2017 to 10 March 2019 over ROIs representative of clean, polluted, and hemispheric regions in order to compare total CO column values from the two instruments. ATom being restricted mostly to oceanic regions precludes the use of this *in situ* dataset for fully validating TROPOMI retrievals over land. To that end, *in situ* data from other airborne measurement programs are required. Ground-

based measurements (e.g., NDACC, TCCON) could also be used; this would allow the validation of seasonal variability at fixed locations. Quantitative comparisons between TROPOMI and MOPITT retrievals over land are relevant, nevertheless. The MOPITT dataset represents the longest global CO record available (2000-present); because of extensive validation efforts with respect to *in situ* measurements and comparisons with other satellite datasets, it is well characterized. Additionally, MOPITT products have served as the reference for many other satellite retrieval products for CO, including AIRS (Worden et al., 2013b), TES (Worden et al., 2013b), and IASI (George et al., 2009, 2015). Furthermore, TROPOMI and MOPITT were, until TANSO-FTS-2 became operational in 2019, the only working satellite instruments retrieving CO from NIR solar-reflected radiances. Thus, it is important to understand their relative behavior, particularly because we are interested in continuing the MOPITT multispectral record (which has enhanced sensitivity to near surface CO for some land observations (Worden et al., 2010)) using radiances from TROPOMI (NIR) and SNPP-CrIS (TIR), two instruments on satellites flying in loose formation (Fu et al., 2016). While our TROPOMI-MOPITT comparisons do not account for the contrasting vertical sensitivities of these two instruments, their results show that there is excellent agreement between the two datasets.

To analyze TROPOMI retrievals over bodies of water, only possible in cloudy conditions, we have used both ATom-4 *in situ* data (24 April to 21 May 2018) and MOPITT TIR retrievals (7 November 2017 to 10 March 2019). The ATom comparison allowed full validation using the TROPOMI AK. This is the ideal situation, since retrieval-simulated ATom-4 column values (i.e., ATom-4 values smoothed using the TROPOMI AK) explicitly account for the TROPOMI retrieval vertical sensitivity (unlike TROPOMI/MOPITT comparisons). The MOPITT comparison provided useful information for a longer period and wider geographical extent, although with the same restrictions noted above regarding the land-only comparisons. Our analyses over bodies of water indicate that TROPOMI's use of reference profiles in cloudy conditions results in errors on the order of a few percent. Since there are no major CO sources over water, CO values closer to the surface (and, therefore, most likely to be below cloud top) tend to be spatially homogeneous and stable through time. Thus, they are well characterized by the reference profiles. (Caution should be exercise in case of sporadic CO sources near open water, e.g., fires near a coastline, which could in some cases result in plumes transported off the coast and below cloud top. Larger errors could occur in such retrievals over water, if sources were not well represented in the TM5 model.) Depending on the representativeness of the TROPOMI reference profiles, larger errors may occur in TROPOMI land retrievals under cloudy conditions, particularly near CO emission sources. These errors require further characterization with colocated *in situ* data and ground measurements over land.

*Data availability.* TROPOMI level 2 CO retrievals for the 7 November 2017 to 27 June 2018 were downloaded from https://s5pexp.copernicus.eu/; retrievals for dates after 28 June 2018 were downloaded from https://s5phub.copernicus.eu/. TROPOMI reference profiles were obtained from ftp://ftp.sron.nl/pub/jochen/TROPOMI_apriori/tm5_co/. MOPITT data can be downloaded from https://doi.org/10.5067/TERRA/MOPITT/MOP02T_L2.008 (TIR), MOP02N_L2.008 (NIR), and MOP02J_L2.008 (TIR+NIR). ATom-4 data from the 7 September 2019 version were downloaded from https://doi.org/10.3334/ORNLDAAC/1581.

*Competing interests.* The authors declare that they have no conflict of interest.

*Acknowledgements.* We thank Clive Rodgers for his insightful comments regarding retrieval methods. NCAR internal reviews provided by Louisa Emmons and John Gille are greatly appreciated. This paper benefited from helpful comments from three anonymous reviewers. This material is based upon work supported by the National Center for Atmospheric Research (NCAR), which is a major facility sponsored by the National Science Foundation under Cooperative Agreement No. 1852977. The NCAR MOPITT project is supported by the National Aeronautics and Space Administration (NASA) Earth Observing System (EOS) Program. This research is supported by NASA-ROSES grant 80NSSC18K0687. Sentinel-5 Precursor is part of the EU Copernicus program, and Copernicus (modified) Sentinel data 2017-2019 has been used. TB is funded through the national TROPOMI program from the NSO. CO measurements on ATom were supported by NASA Earth Venture program through grants NNX15AJ23G to Harvard University and NNX16AL92A to the University of Colorado.

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

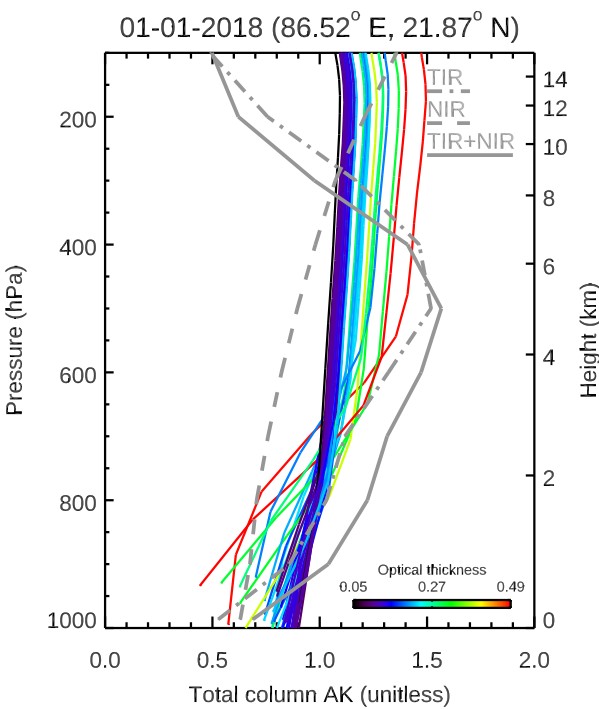

**Figure 1.** Total column AK (averaging kernels) from MOPITT and TROPOMI observations acquired 1 January 2018. Gray lines show AK from a single clear MOPITT pixel. Color-coded lines show AK from TROPOMI observations colocated with that MOPITT pixel (same day acquisition, $\leq$ 50 km horizontal distance) with optical depth <0.5 and cloud height <5000 m (i.e., clear-sky, clear-sky-like, and mid-level-cloud observations). Differences in TROPOMI AK vertical extent are due to topography.

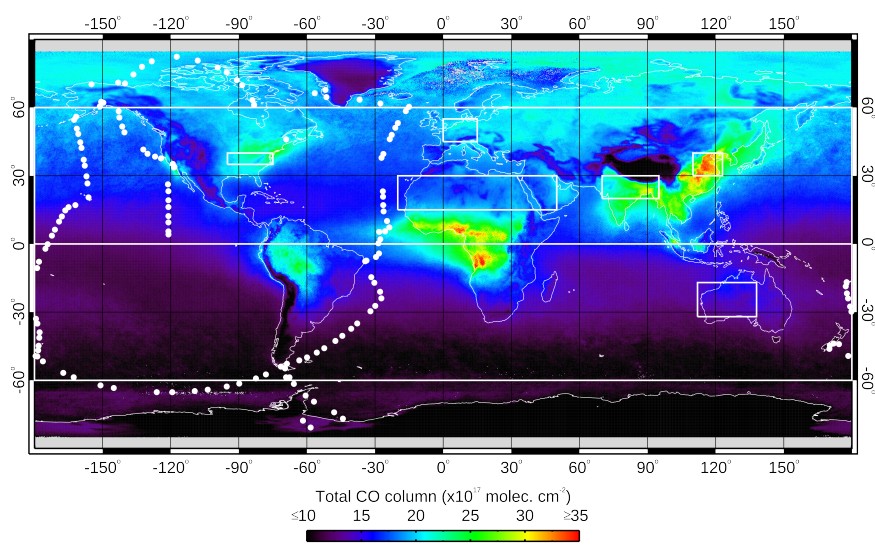

**Figure 2.** White rectangles show the location of land-only ROIs analyzed: N Hemisphere (0°N to 60°N), S Hemisphere (60°S to 0°N), USA (35°N, 95°W to 40°N, 75°W), Europe (45°N, 0°E to 55°N, 15°E), India (20°N, 70°E to 30°N, 95°E), China (30°N, 110°E to 40°N, 123°E), Sahara (15°N, 20°W to 30°N, 50°E), and Australia (32°S, 112°E to 17°S, 138°E). White circles indicate location of individual CO profiles acquired in April-May 2018, during the ATom-4 airborne campaign. Background map shows mean MOPITT TIR total CO column values for 2018.

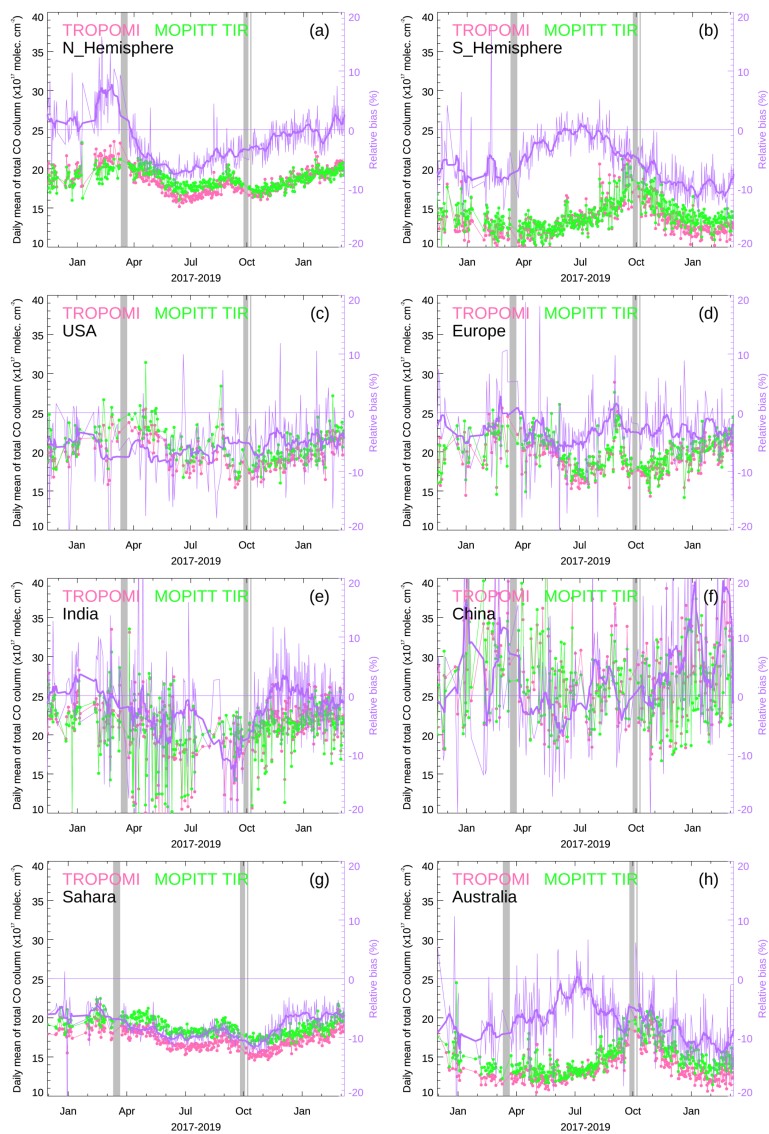

**Figure 3.** Comparison of colocated land retrievals from TROPOMI (pink) and MOPITT TIR (green) for each ROI analyzed. Filled circles show daily mean. Thin purple lines indicate daily relative bias (i.e., accuracy) between the two datasets, thick purple lines are a 11-day smoothed version with high-frequency variability removed. Gray bars show periods without MOPITT measurements because of hot calibrations (March and October 2018) or a safe mode maneuver (October-November 2018).

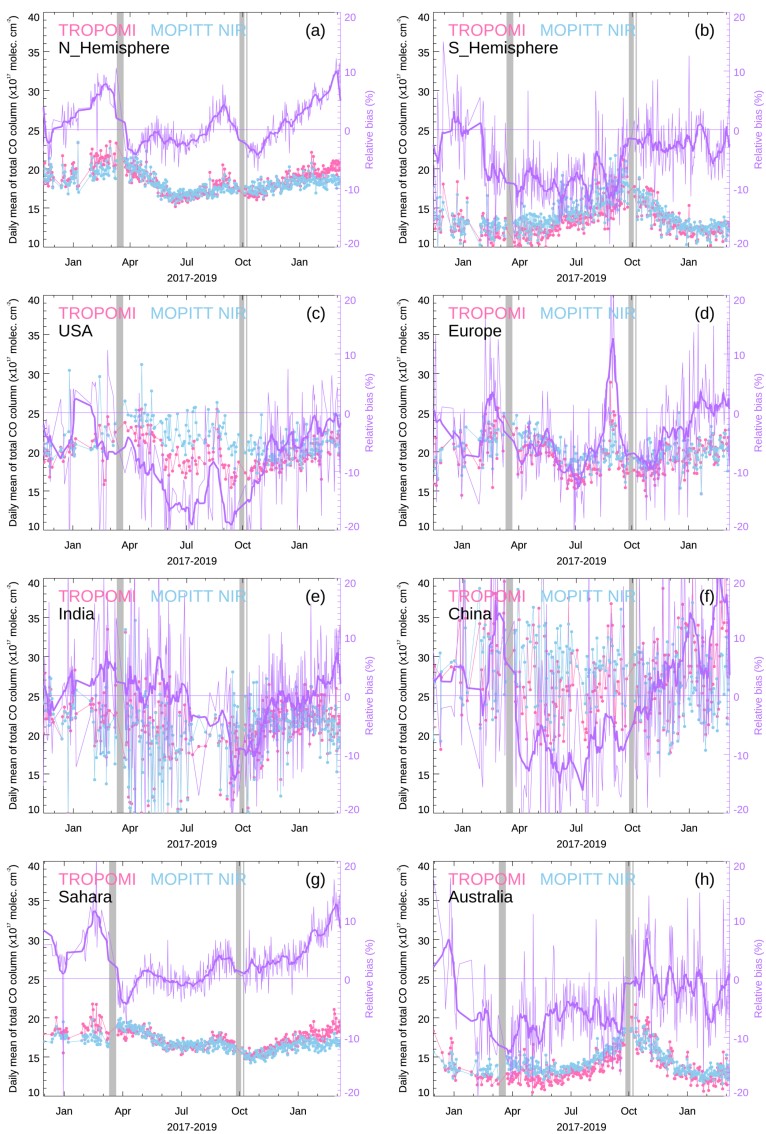

**Figure 4.** Comparison of colocated land retrievals from TROPOMI (pink) and MOPITT NIR (blue) for each ROI analyzed. See caption to Fig. 3 for details.

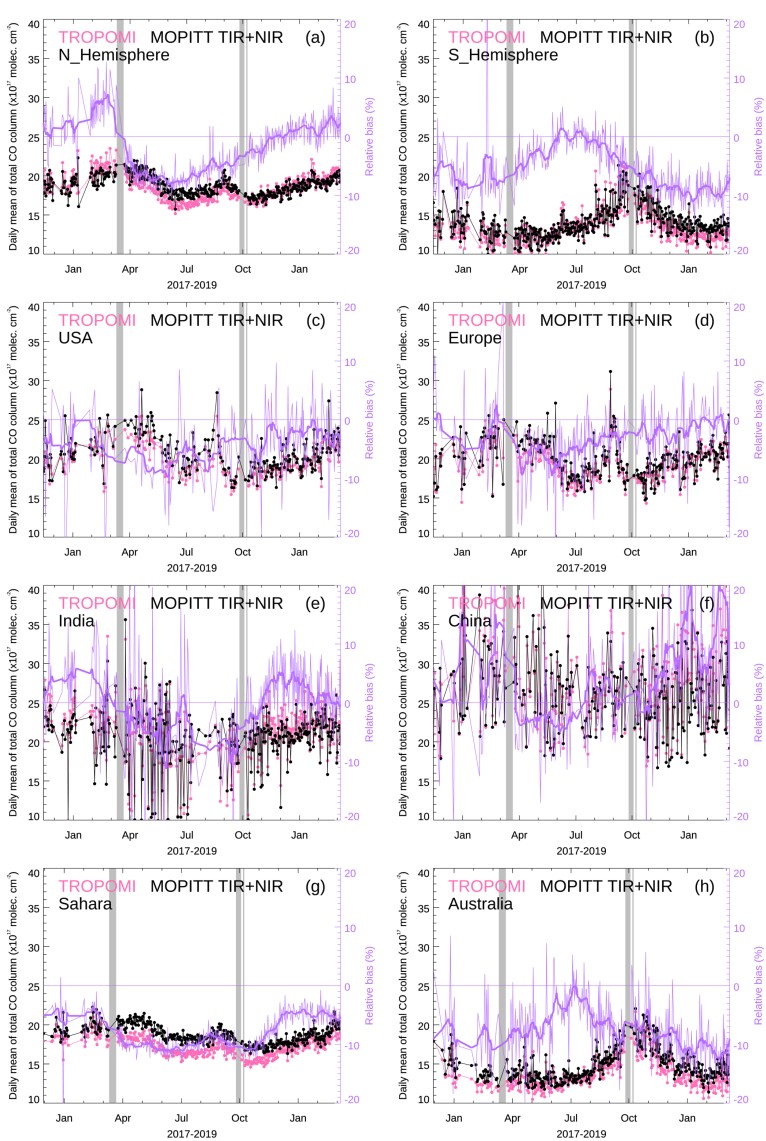

**Figure 5.** Comparison of colocated land retrievals from TROPOMI (pink) and MOPITT TIR+NIR (black) for each ROI analyzed. See caption to Fig. 3 for details.

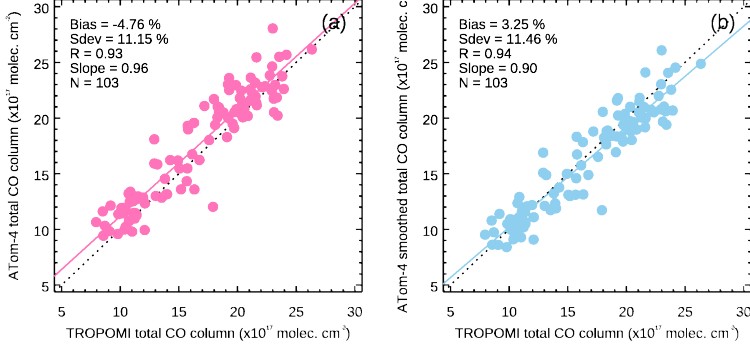

**Figure 6.** Comparison of colocated retrievals over bodies of water from TROPOMI and ATom-4 (24 April - 21 May 2018). a) TROPOMI versus true (unsmoothed) ATom-4. b) TROPOMI versus retrieval-simulated (smoothed) ATom-4.

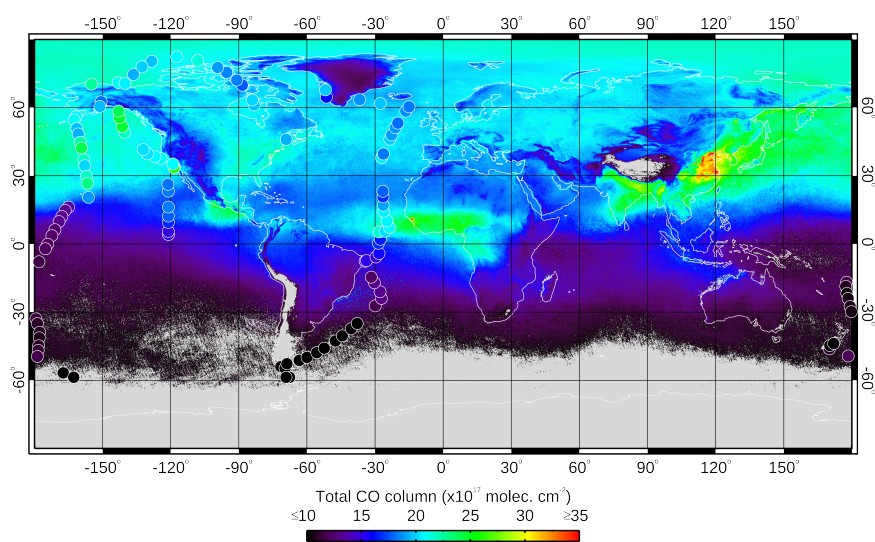

**Figure 7.** Map of averaged TROPOMI total CO column values acquired between 24 April and 21 May 2018, the duration of the ATom-4 campaign. Circles show ATom-4 profiles spatially and temporally colocated with single TROPOMI retrievals; circles are color-coded according to their retrieval-simulated (smoothed) ATom total CO column value. There is good agreement between the two datasets, despite differences in the time span and footprint size each of them represents.

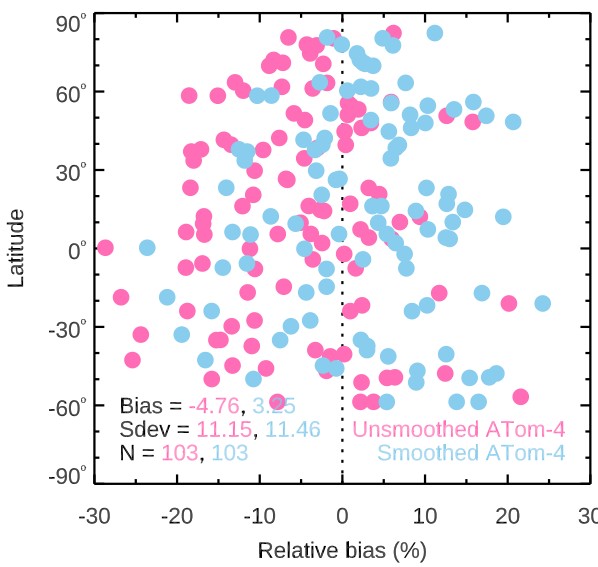

**Figure 8.** Latitudinal distribution of relative bias between TROPOMI and ATom-4 over bodies of water. Negative bias indicates that TROPOMI retrievals are low with respect to ATom-4.

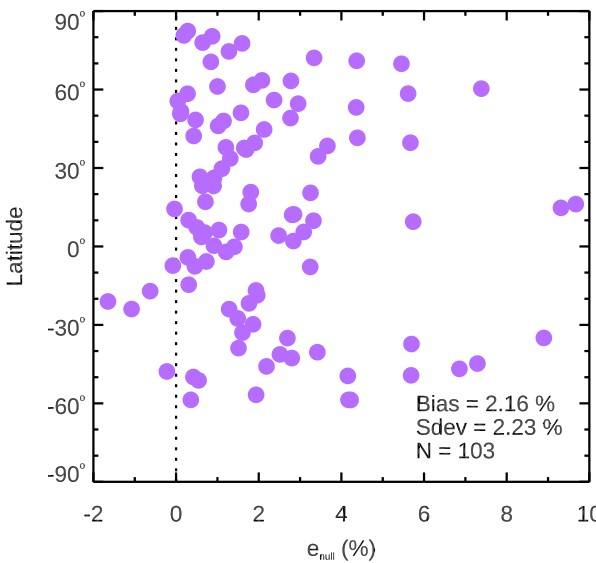

**Figure 9.** Latitudinal distribution of $e_{null}$ error (see Eq. (4)), which characterizes retrieval errors over cloudy scenes related to the shape of the TROPOMI model-calculated reference profiles, expressed in percentage with respect to the true (unsmoothed) ATom-4 total CO columns.

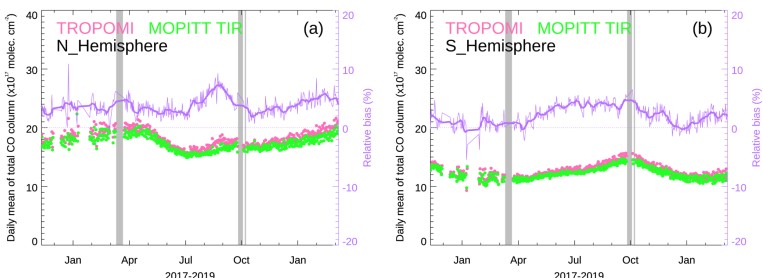

**Figure 10.** Comparison of colocated retrievals over bodies of water from TROPOMI and MOPITT TIR. a) Compilation of means and relative biases of total CO column values from 7 November 2017 to 10 March 2019 for the N Hemisphere ROI. b) Same for the S Hemisphere ROI.

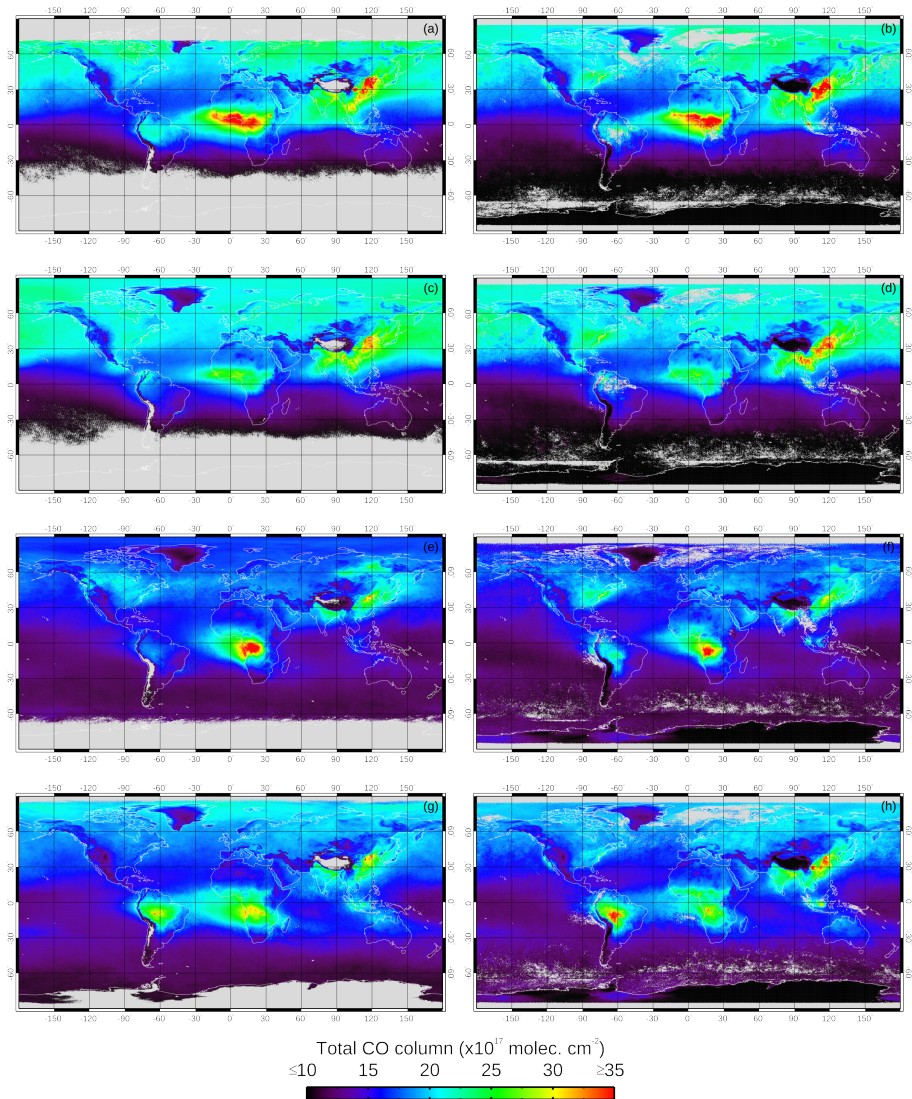

**Figure 11.** Seasonal averages of TROPOMI and MOPITT TIR CO retrievals. a) December 2017 to February 2018 (DJF) TROPOMI mean. b) Same for MOPITT. c) March-May 2018 (MAM) TROPOMI mean. d) Same for MOPITT. e) June-August 2018 (JJA) TROPOMI mean. f) Same for MOPITT. g) September-November 2018 (SON) TROPOMI mean. h) Same for MOPITT. Sharp discontinuities visible in some panels at 65°S are due to differences in the definition of the MOPITT cloud mask poleward of latitude 65°.

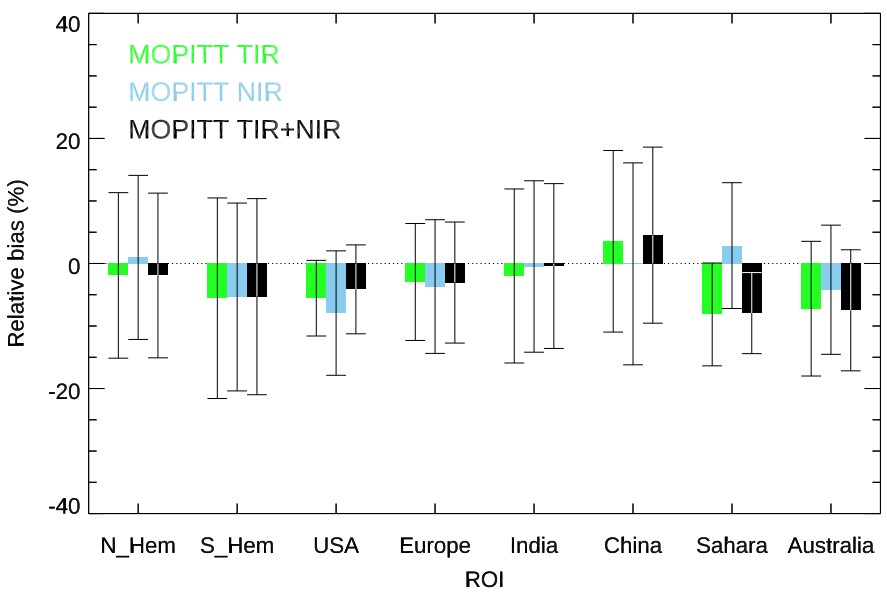

**Figure 12.** Summary of colocated land comparison results. Colored bars represent relative bias between TROPOMI and each of the three MOPITT products (TIR, NIR, and TIR+NIR); solid lines indicate the standard deviation of relative bias

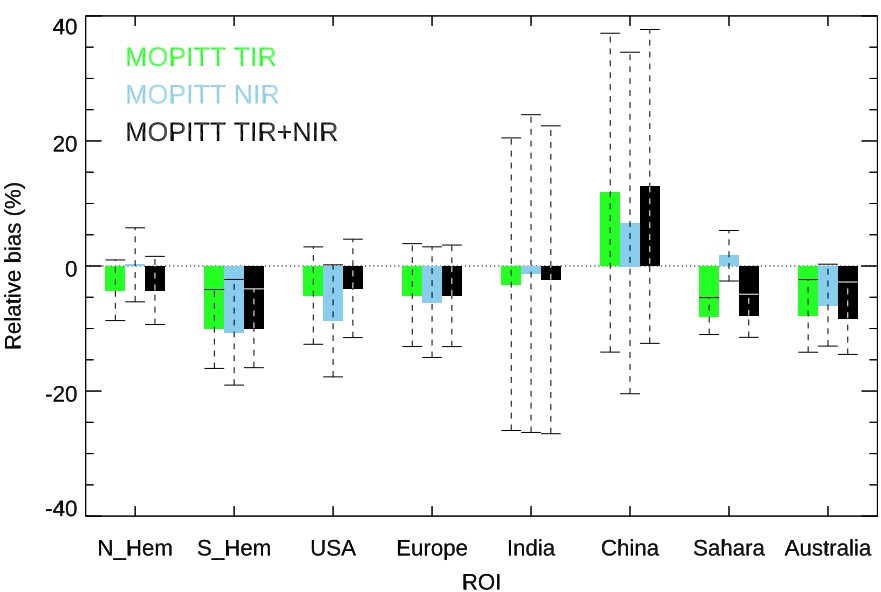

**Figure 13.** Summary of non-colocated land comparison results. Colored bars represent relative bias between TROPOMI and each of the three MOPITT products (TIR, NIR, and TIR+NIR). Dashed lines show $\pm$ 1 standard deviation of mean daily relative biases (i.e., inter-daily bias variability).

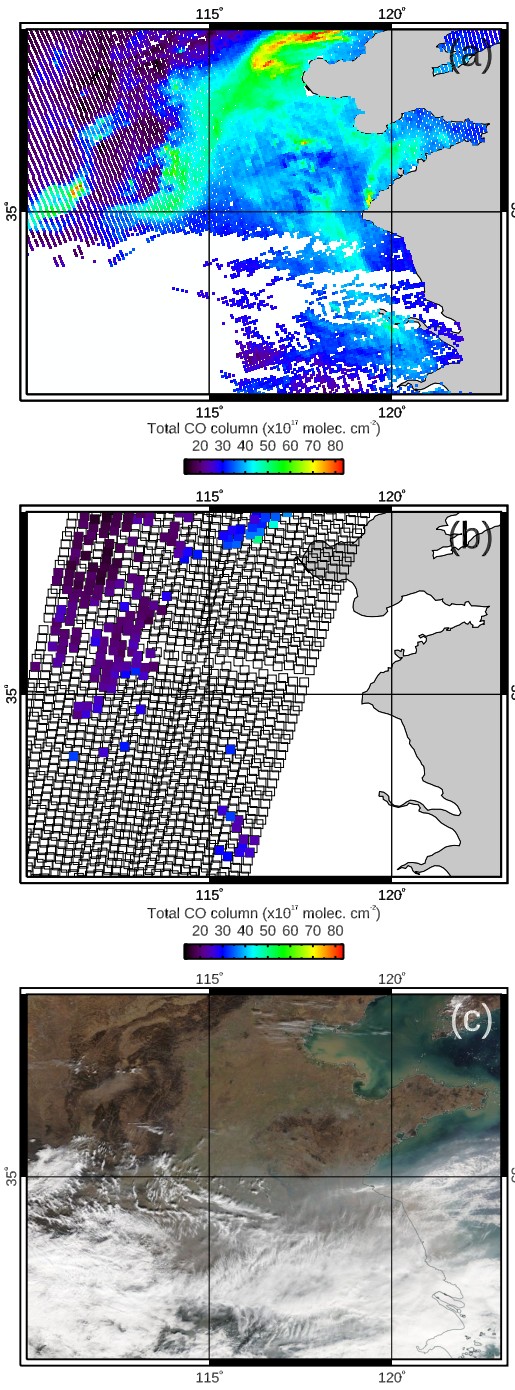

**Figure 14.** Total CO column retrievals and visible image for the China ROI on 1 January 2018. a) TROPOMI map. b) MOPITT TIR+NIR map. c) Terra-MODIS visible image acquired at the same time as the MOPITT data. Empty boxes in the second panel correspond to MOPITT observations deemed cloudy based on MODIS cloud mask information, and thus not suitable for CO retrieval. The MODIS visible image shows clouds in the southern half of the ROI; the northern half was hazy, most probably due to pollution, but cloud-free.

**Table 1.** Statistics from colocated TROPOMI versus MOPITT CO retrievals over land for the period between 7 November 2017 and 10 March 2019. Relative bias and standard deviation in %. Column bias and standard deviation in units of $10^{17}$ molec. cm$^{-2}$.

| | | TROPOMI vs MOPITT$_{TIR}$ | TROPOMI vs MOPITT$_{NIR}$ | TROPOMI vs MOPITT$_{TIR+NIR}$ |
|---|---|---|---|---|
| N Hemisphere | Relative Bias±St. Dev. | -1.91±13.24 | 0.97±13.12 | -1.92±13.17 |
| | Column Bias±St. Dev. | -0.55±2.51 | -0.04±2.58 | -0.55±2.45 |
| | Mean Daily Colocated Pairs | 45672 | 45678 | 45530 |
| S Hemisphere | Relative Bias±St. Dev. | -5.56±16.04 | -5.36±15.02 | -5.31±15.68 |
| | Column Bias±St. Dev. | -1.02±2.50 | -0.95±2.32 | -0.95±2.30 |
| | Mean Daily Colocated Pairs | 7768 | 7771 | 7748 |
| USA | Relative Bias±St. Dev. | -5.55±6.05 | -7.93±9.95 | -4.14±7.11 |
| | Column Bias±St. Dev. | -1.25±1.33 | -2.02±2.36 | -1.00±1.53 |
| | Mean Daily Colocated Pairs | 666 | 686 | 666 |
| Europe | Relative Bias±St. Dev. | -2.96±9.35 | -3.69±10.69 | -3.05±9.68 |
| | Column Bias±St. Dev. | -0.73±1.84 | -0.91±2.29 | -0.79±2.04 |
| | Mean Daily Colocated Pairs | 657 | 661 | 656 |
| India | Relative Bias±St. Dev. | -2.00±13.92 | -0.48±13.71 | -0.41±13.18 |
| | Column Bias±St. Dev. | -0.74±2.80 | -0.47±2.90 | -0.38±2.43 |
| | Mean Daily Colocated Pairs | 1122 | 1133 | 1118 |
| China | Relative Bias±St. Dev. | 3.55±14.52 | -0.06±16.15 | 4.53±14.08 |
| | Column Bias±St. Dev. | 0.74±4.00 | -0.37±4.64 | 0.98±3.86 |
| | Mean Daily Colocated Pairs | 533 | 566 | 534 |
| Sahara | Relative Bias±St. Dev. | -8.15±8.22 | 2.86±10.06 | -7.94±6.48 |
| | Column Bias±St. Dev. | -1.64±1.64 | 0.34±1.72 | -1.55±1.27 |
| | Mean Daily Colocated Pairs | 15214 | 15223 | 15169 |
| Australia | Relative Bias±St. Dev. | -7.23±10.77 | -4.20±10.33 | -7.49±9.68 |
| | Column Bias±St. Dev. | -1.28±1.85 | -0.69±1.52 | -1.26±1.57 |
| | Mean Daily Colocated Pairs | 1873 | 1875 | 1869 |
| Mean all ROIs | Relative Bias±St. Dev. | -3.73±11.51 | -2.24±12.38 | -3.22±11.13 |
| | Column Bias±St. Dev. | -0.81±2.31 | -0.64±2.54 | -0.69±2.18 |
| | Mean Daily Colocated Pairs | 9188 | 9199 | 9161 |

**Table 2.** Colocated TROPOMI versus ATom-4 CO retrievals over bodies of water: Statistics from AK analysis. Relative bias and standard deviation in %. Column bias and standard deviation in units of $10^{17}$ molec. $cm^{-2}$.

| | | TROPOMI vs True ATom-4 (Unsmoothed) | TROPOMI vs Retrieval-Simulated ATom-4 (Smoothed) |
|---|---|---|---|
| Atlantic/Pacific | Relative Bias±St. Dev. | -4.76±11.15 | 3.25±11.46 |
| | Column Bias±St. Dev. | -0.89±1.80 | 0.46±1.68 |
| | Number of Colocated Pairs | 103 | 103 |

**Table 3.** Colocated TROPOMI versus MOPITT TIR CO retrievals over bodies of water: Statistics performed for the period between 7 November 2017 and 10 March 2019. Relative bias and standard deviation in %. Column bias and standard deviation in units of $10^{17}$ molec. cm$^{-2}$.

| | | TROPOMI vs MOPITT$_{TIR}$ Total Column |
|---|---|---|
| N Hemisphere | Relative Bias±St. Dev. | 3.82±13.27 |
| | Column Bias±St. Dev. | 0.53±2.35 |
| | Mean Daily Colocated Pairs | 127360 |
| S Hemisphere | Relative Bias±St. Dev. | 2.14±18.15 |
| | Column Bias±St. Dev. | 0.19±2.38 |
| | Mean Daily Colocated Pairs | 164935 |
| Mean both Hemispheres | Relative Bias, St. Dev. | 2.98±15.71 |
| | Column Bias±St. Dev. | 0.36±2.37 |
| | Mean Daily Colocated Pairs | 146148 |

**Table 4.** Statistics from colocated, null-space adjusted TROPOMI versus MOPITT CO retrievals over land for the period between 7 November 2017 and 10 March 2019. Relative bias and standard deviation in %. Column bias and standard deviation in units of $10^{17}$ molec. cm$^{-2}$.

| | | TROPOMI vs MOPITT$_{TIR}$ | TROPOMI vs MOPITT$_{NIR}$ | TROPOMI vs MOPITT$_{TIR+NIR}$ |
|---|---|---|---|---|
| N Hemisphere | Relative Bias±St. Dev. | -1.19±13.31 | 1.68±13.05 | -1.19±13.28 |
| | Column Bias±St. Dev. | -0.40±2.52 | 0.10±2.55 | -0.41±2.47 |
| | Mean Daily Colocated Pairs | 45672 | 45678 | 45530 |
| S Hemisphere | Relative Bias±St. Dev. | -4.74±16.08 | -4.60±14.90 | -4.48±15.76 |
| | Column Bias±St. Dev. | -0.91±2.49 | -0.83±2.28 | -0.84±2.30 |
| | Mean Daily Colocated Pairs | 7768 | 7771 | 7748 |
| USA | Relative Bias±St. Dev. | -2.62±6.21 | -5.12±10.19 | -1.17±7.30 |
| | Column Bias±St. Dev. | -0.65±1.34 | -1.42±2.36 | -0.40±1.54 |
| | Mean Daily Colocated Pairs | 666 | 686 | 666 |
| Europe | Relative Bias±St. Dev. | -0.97±9.49 | -1.72±10.88 | -1.05±9.85 |
| | Column Bias±St. Dev. | -0.34±1.85 | -0.52±2.30 | -0.39±2.05 |
| | Mean Daily Colocated Pairs | 657 | 661 | 656 |
| India | Relative Bias±St. Dev. | -0.95±13.84 | 0.52±13.59 | 0.68±13.13 |
| | Column Bias±St. Dev. | -0.48±2.79 | -0.21±2.85 | -0.11±2.44 |
| | Mean Daily Colocated Pairs | 1122 | 1133 | 1118 |
| China | Relative Bias±St. Dev. | 5.44±14.59 | 1.77±16.19 | 6.44±14.17 |
| | Column Bias±St. Dev. | 1.25±4.00 | 0.16±4.61 | 1.49±3.86 |
| | Mean Daily Colocated Pairs | 533 | 566 | 534 |
| Sahara | Relative Bias±St. Dev. | -8.00±8.24 | 3.02±10.05 | -7.79±6.50 |
| | Column Bias±St. Dev. | -1.61±1.64 | 0.37±1.71 | -1.52±1.27 |
| | Mean Daily Colocated Pairs | 15214 | 15223 | 15169 |
| Australia | Relative Bias±St. Dev. | -7.13±10.76 | -4.11±10.32 | -7.39±9.67 |
| | Column Bias±St. Dev. | -1.27±1.85 | -0.68±1.52 | -1.25±1.57 |
| | Mean Daily Colocated Pairs | 1873 | 1875 | 1869 |
| Mean all ROIs | Relative Bias±St. Dev. | -2.52±11.57 | -1.07±12.40 | -1.99±11.21 |
| | Column Bias±St. Dev. | -0.55±2.31 | -0.38±2.52 | -0.43±2.19 |
| | Mean Daily Colocated Pairs | 9188 | 9199 | 9161 |

**Table 5.** Colocated, null-space adjusted TROPOMI versus MOPITT TIR CO retrievals over bodies of water: Statistics analysis performed for the period between 7 November 2017 and 10 March 2019. Relative bias and standard deviation in %. Column bias and standard deviation in units of $10^{17}$ molec. cm$^{-2}$.

| | | TROPOMI vs MOPITT$_{TIR}$ Total Column |
|---|---|---|
| N Hemisphere | Relative Bias±St. Dev. | 5.90±13.19 |
| | Column Bias±St. Dev. | 0.91±2.32 |
| | Mean Daily Colocated Pairs | 127360 |
| S Hemisphere | Relative Bias±St. Dev. | 3.82±18.11 |
| | Column Bias±St. Dev. | 0.39±2.36 |
| | Mean Daily Colocated Pairs | 164544 |
| Mean both Hemispheres | Relative Bias, St. Dev. | 4.86±15.65 |
| | Column Bias±St. Dev. | 0.65±2.34 |
| | Mean Daily Colocated Pairs | 145952 |