# Peer review of "1.5 years of TROPOMI CO measurements: Comparisons to MOPITT and ATom"

_Atmospheric Measurement Techniques, 2020_

## Referee Comment (RC1) · Anonymous Referee #1 · 6 Apr 2020

General Comments:

The authors provided a validation for the TROPOMI CO observations by using MOPITT and Atom CO measurements. I found the paper is well written. The authors demonstrated good agreement between TROPOMI and MOPITT CO observations, which is helpful for people who are interested in the sources and variations of CO from global to regional scales. I recommend the paper for publication after consideration of the minor points below.

Specific Comments:

1. Lines 154-160: It is difficult to follow this paragraph. I checked Section 2 again but didn't find the details to support the direct comparison without transformation. It would

be better to provide more details here.

2. Section 4.1. The comparison between TROPOMI and MOPITT is very interesting. I have some suggestions, which may be considered in this work or the following study: 1) The differences between two datasets show obvious seasonal variabilities. What are the possible explanations? 2) We know that MOPITT show some latitude-depended differences relative to surface and aircraft measurements. Do the differences between TROPOMI and MOPITT have similar latitudinal dependence?

---

## Referee Comment (RC2) · Anonymous Referee #2 · 6 Apr 2020

This manuscript compares TROPOMI carbon monoxide retrievals to data from the MO-PITT satellite and in situ airborne profiles (ATom-4). The manuscript is well written and falls into the scope of AMT. I recommend publication after the following comments have been addressed.

**General Comments**

My main concern is that the significantly different vertical sensitivities of the instruments and the different apriori profiles used in the algorithms are not taken into account in the comparison of the TROPOMI and MOPITT data. It is alleged that the corresponding

comparison methodology is not applicable to profile scaling retrievals. However, I do not agree with this view as TROPOMI's averaging kernels (AKs) take into account that it is a profile scaling retrieval. The AK value of the i-th layer quantifies as usual the sensitivity of the total column to a change of CO in the i-th layer. It is also not a question of constraining the results with the apriori or not. If the AKs are not a direct output of the retrieval, you can simply compute them for every kind of algorithm by confronting the retrieval with simulated measurements and doing the following for each layer i: 1) change the abundance in the i-th layer, 2) perform the retrieval, 3) compare the retrieved column to the "true" column. For a meaningful comparison, at least the individual apriori profiles of both retrievals should be replaced by a common prior by using the AKs (see e.g. Section 4 of Rodgers and Connor (2003) or Appendix A of Wunch et al. (2011)). The common prior can be the TROPOMI prior, the MOPITT prior, or a different third prior. Please improve the comparison method by taking these aspects into account or give a justification why the consideration of the AKs is negligible in this analysis and prove by example that the figures of merit like the global bias between the two data sets do not critically depend on whether the individual apriori profiles are replaced by a common prior.

**Specific Comments**

Page 1, Lines 4-5: TANSO-FTS-2 on the GOSAT-2 satellite (launched in 2018) is also deriving CO from solar reflected radiances in the $2.3\,\mu$m spectral region.

Page 1, Lines 16-17: see general comments

Page 3, Lines 49-50: TANSO-FTS-2 on the GOSAT-2 satellite is also deriving CO from solar reflected radiances in the $2.3\,\mu$m spectral region.

Page 7, Lines 157-160: see general comments

Page 8, Line 195: The units in Eq. (1) do not match.

Page 9, Lines 223-225: Why not use the actual TROPOMI averaging kernels here instead of a binary step function?

Page 10, Lines 242-243: Why not use the actual TROPOMI averaging kernels?

Page 13, Lines 342-343: The negative bias could simply be a consequence of the different sensitivities and apriori profiles used in the estimation of the true atmospheric state for the two individual instruments. Thus, consideration of the averaging kernels is important. Please note for example the change in sign for the biases in Figure 6 due to the AKs.

Page 14, Lines 378-380: How do these error estimates change when considering the averaging kernels and apriori profiles?

Page 14, Lines 384-386: In addition to the global accuracy and precision, it would also be interesting to quantify the regional relative accuracy quantifying region-to-region biases, e.g. the standard deviation of the individual biases for the regions of Figures 2-5. The precision requirement of 10 % is not satisfied.

Page 14, Line 390: To validate TROPOMI retrievals over land, ground-based measurements from the Total Carbon Column Observing Network (TCCON), which are calibrated using aircraft profiles, can also be used. In contrast to aircraft data this would also allow the validation of seasonal variability at a fixed location.

Page 14, Line 392-393: I would call a comparison to other satellite data sets verification instead of validation. Has the seasonal variability of MOPITT also been validated, e.g. by using TCCON or NDACC ground-based measurements?

Page 14, Line 395-396: TANSO-FTS-2 on the GOSAT-2 satellite is also deriving CO from solar reflected radiances in the $2.3\,\mu$m spectral region.

Page 14, Line 399: Please replace "do not fully account for" by "do not account for".

Page 15, Line 408: What about transport of CO from major sources in coastal regions to the ocean?

Page 15, Line 411-412: or validation with ground-based measurements (TCCON or NDACC)

Figures 3-5: Please add regional mean bias and standard deviation of the differences to all individual subplots.

Figures 13-14: Please add the standard deviation of the individual regional biases as a measure of the region-to-region bias to the plots (for TIR, NIR, and TIR+NIR).

Table 1: Please add region-to-region biases in case "all ROIs".

**Technical Corrections**

Page 3, Line 50: Please rephrase "is key."

Page 8, Line 195: Please replace $\Sigma_{i=1}^{i=n}$ by $\Sigma_{i=1}^{n}$ in Eq.(1).

Page 13, Line 340: Please add the unit % for the relative biases.

**References**

Rodgers, C. D. and Connor, B. J.: Intercomparison of remote sounding instruments, J. Geophys. Res., 108, D3, 4116, https://doi.org/10.1029/2002JD002299, 2003.

Wunch, D., Wennberg, P. O., Toon, G. C., Connor, B. J., Fisher, B., Osterman, G. B., Frankenberg, C., Mandrake, L., O'Dell, C., Ahonen, P., Biraud, S. C., Castano, R., Cressie, N., Crisp, D., Deutscher, N. M., Eldering, A., Fisher, M. L., Griffith, D. W. T., Gunson, M., Heikkinen, P., Keppel-Aleks, G., Kyro, E., Lindenmaier, R., Macatangay,

[Figure]

R., Mendonca, J., Messerschmidt, J., Miller, C. E., Morino, I., Notholt, J., Oyafuso, F. A., Rettinger, M., Robinson, J., Roehl, C. M., Salawitch, R. J., Sherlock, V., Strong, K., Sussmann, R., Tanaka, T., Thompson, D. R., Uchino, O., Warneke, T., and Wofsy, S. C.: A method for evaluating bias in global measurements of $CO_2$ total columns from space, Atmos. Chem. Phys., 11, 12317-12337, https://doi.org/10.5194/acp-11-12317-2011, 2011.

---

## Referee Comment (RC3) · Anonymous Referee #3 · 2 May 2020

The authors have conducted a validation of TROPOMI CO retrievals using data from MOPITT and aircraft profiles of CO from ATom. The TROPOMI data are fairly new and provide tremendous observational coverage at high spatial resolution. However, MOPITT offers a uniquely long record of space-based measurements of CO, therefore there is significant value in the validation analysis presented here. My main concern is that when comparing two remote sensing data sets it is critical to account for the influence of the a priori profiles on the retrievals and for the different vertical sensitivities of the measurements, which was not done in this study. The manuscript is well written and appropriate for AMT. I would recommend publication of the manuscript after the authors have addressed my comments below.

General Comments

[Figure]

1. Lines 111-114: The discussion here is somewhat confusing. The authors state that the NIR retrievals are significantly constrained by the a priori, whereas the TIR are less strongly weighted by the a priori profile. However, on lines 159-160 they explain that they do not transform the MOPITT and TROPOMI profiles when comparing them. It would seem that the different contributions of the a priori to the two sets of retrievals would necessitate accounting for the influence of the a priori profiles to meaningfully compare the two data sets. What is the justification for neglecting this?

2. Lines 162-163: What is the impact of the differences in the overpass times of TROPOMI and MOPITT when selecting "collocated" pairs of data? Quantifying this for the ROIs selected in the study would be helpful for interpreting the results of the intercomparison.

3. Lines 307-308: What is the implication of the tendency of the reference profiles to have too much CO near the surface for the intercomparison with MOPITT, considering that no attempt is made to mitigate potential biases arising from the a priori?

4. Lines 354-357: It is certainly possible that the differences in overpass times could contribute to these biases over Africa, but this can be confirmed with a model, for example. Furthermore, what about the impact of the different vertical sensitivities of the measurements here? It seems critical to me to account for the influence of the averaging kernels before speculating that these differences could be due to temporal variations in the African fires.

Technical Comments

1) Line 46: This is not the first use of the acronym MOPITT.

2) Line 110: Please insert "the" before "total column AK".

3) Line 119: Please make it clear that "(∼480; note 1 km resolution)" here is referring to the number of MODIS observation, and that these observations have a resolution of 1 km.

---

## Author Comment (AC3) · 22 Jul 2020

The authors have conducted a validation of TROPOMI CO retrievals using data from MOPITT and aircraft profiles of CO from ATom. The TROPOMI data are fairly new and provide tremendous observational coverage at high spatial resolution. However, MOPITT offers a uniquely long record of space-based measurements of CO, therefore there is significant value in the validation analysis presented here. My main concern is that when comparing two remote sensing data sets it is critical to account for the influence of the a priori profiles on the retrievals and for the different vertical sensitivities of the measurements, which was not done in this study. The manuscript is well written and appropriate for AMT. I would recommend publication of the manuscript after the authors have addressed my comments below.

General Comments
1. Lines 111-114: The discussion here is somewhat confusing. The authors state that the NIR retrievals are significantly constrained by the a priori, whereas the TIR are less strongly weighted by the a priori profile. However, on lines 159-160 they explain that they do not transform the MOPITT and TROPOMI profiles when comparing them. It would seem that the different contributions of the a priori to the two sets of retrievals would necessitate accounting for the influence of the a priori profiles to meaningfully compare the two data sets. What is the justification for neglecting this?

Thank you for this comment. New text and two tables with results from an additional analysis have been included in the manuscript to 1) better justify the direct comparisons without transformation and 2) investigate the effect on biases of the differences between MOPITT *a priori* CO profiles and TROPOMI reference CO profiles. New Section 3.1 discusses in more detail the differences between the MOPITT and TROPOMI CO retrieval algorithms, as well as the challenges these differences impose when comparing the two datasets. New Section 3.3.1 discusses the main sources of error in satellite CO retrievals; it also discusses sources of error when comparing satellite datasets, e.g., differences in *a priori* information used by each dataset and differences in vertical sensitivity (represented by the averaging kernels, or AKs) between instruments.
Determining whether or not observed differences in retrievals from these two instruments are consistent with differences in their *a priori*, AKs, and instrument noise would require knowledge of the true atmosphere during observation; this information is often unavailable, here included. Our main goal in comparing MOPITT and TROPOMI total CO column retrievals is to quantify differences between the two retrieval products available to users, rather than quantify the actual bias of either product. This goal is addressed by direct "end to end" comparisons of the two untransformed products in various regions of interest, after colocation of the MOPITT and TROPOMI retrievals. These comparisons quantify the MOPITT/TROPOMI difference statistics due to all effects: AK differences, *a priori* differences, and instrument noise.
Additionally, we now investigate the effects of differences between the *a priori*/reference information used by MOPITT and TROPOMI in their retrievals; we do so by applying a null-space adjustment (based on the MOPITT *a priori*) to TROPOMI. We present results from this additional analysis in

Sections 4.1.4 and 4.2.3 and show that differences in *a priori*/reference CO profiles affect MOPITT/TROPOMI relative biases by 1-2 percentage points, well below TROPOMI's required 15% accuracy.

2. Lines 162-163: What is the impact of the differences in the overpass times of TROPOMI and MOPITT when selecting "collocated" pairs of data? Quantifying this for the ROIs selected in the study would be helpful for interpreting the results of the intercomparison. Quantifying the effect of differences in passing times in CO retrievals is an interesting topic, but it is outside the scope of this work. Please note that validation papers allow time differences substantially larger than the 3 hours between MOPITT and TROPOMI, e.g., Deeter et al., 2019 (12 hours); Clerbaux et al, 2008 (24 hours). The lifetime of CO (several weeks) is much greater than the time difference between MOPITT and TROPOMI passing times. Differences in total CO column amounts due to transportation would be equally likely to be positive or negative; thus, they would not contribute to an apparent bias between the two products.

3. Lines 307-308: What is the implication of the tendency of the reference profiles to have too much CO near the surface for the intercomparison with MOPITT, considering that no attempt is made to mitigate potential biases arising from the a priori? Thank you for this comment. Please see response to Comment #1 above. As explained there, the manuscript now includes text describing (and results from) an additional analysis where we quantify the effect on biases of the differences between MOPITT *a priori* CO profiles and TROPOMI reference CO profiles. We show that differences in *a priori*/reference CO profiles affect MOPITT/TROPOMI relative biases by 1-2 percentage points, well below TROPOMI's required 15% accuracy.

4. Lines 354-357: It is certainly possible that the differences in overpass times could contribute to these biases over Africa, but this can be confirmed with a model, for example. Modeling the effect of differences in passing times in CO retrievals is an interesting topic, but it is outside the scope of this work. Also, please note that lines 354-357 discuss results obtained for the China ROI; furthermore, please note that the ROIs analyzed in this work do not include fire regions in Africa. Furthermore, what about the impact of the different vertical sensitivities of the measurements here? It seems critical to me to account for the influence of the averaging kernels before speculating that these differences could be due to temporal variations in the African fires. Thank you for this comment. Please see response to Comment #1 for more details regarding additional text now included in the manuscript to address this point.

Technical Comments

1) Line 46: This is not the first use of the acronym MOPITT. Thank you for catching this. We have reworded lines 33-35 to include definitions of TROPOMI, MOPITT, and ATom the first time they are mentioned in the Introduction: "The aim of this work is to facilitate the extension of the current satellite record with newly available TROPOMI (TROPOspheric Monitoring Instrument) measurements by evaluating those with respect to satellite MOPITT (Measurements Of Pollution In The Troposphere) and *in situ* ATom (Atmospheric Tomography mission) CO data." Also, the MOPITT and ATom acronym definitions in lines 46-47 have been removed and the sentence reworded to: "Here we analyze daily global TROPOMI retrievals acquired between 7 November 2017 and 10 March 2019 with respect to MOPITT and ATom."

2) Line 110: Please insert "the" before "total column AK". Please note line 110 does not contain the text "total column AK". That text appears, though, in lines 109 and 111; we have added "the" to the

latter occurrence. That sentence now reads: "With respect to vertical sensitivity, the total column AK for the NIR-only product are most similar in shape to the TROPOMI total column AK"

3) Line 119: Please make it clear that "(∼480; note 1 km resolution)" here is referring to the number of MODIS observation, and that these observations have a resolution of 1 km. Thank you for this comment. For increased clarity that sentence has been reworded to: "The ~480 MODIS observations at $1 \times 1 \text{ km}^2$ horizontal resolution acquired at the same time as a single MOPITT observation and within the MOPITT footprint are identified and collected"

---

## Author Response (AR1)

This document contains:
- Responses to Review #1
- Responses to Review #2
- Responses to Review #3
- Marked-up file (*) showing differences between the submitted manuscript and the revised manuscript.

To address overlapping comments from all three reviews, we have performed an additional analysis to quantify bias produced by differences between MOPITT *a priori* CO profiles and TROPOMI reference CO profiles. We did so by applying a null-space adjustment (based on the MOPITT *a priori*) to TROPOMI. Our results show that bias values change by only 1-2 percentage points respect to the original bias values.

In summary: we have added a new Section 3.1 discussing in more detail the differences between the MOPITT and TROPOMI CO retrieval algorithms, as well as the challenges these differences impose when comparing the two datasets. New Section 3.3.1 discusses the main sources of error in satellite CO retrievals and sources of error when comparing satellite datasets (i.e., differences in *a priori* and AKs). New Sections 4.1.4 and 4.2.3, and new Tables 4 and 5, summarize biases between MOPITT and null-space adjusted TROPOMI over land and ocean. For focus, we have moved the above/below cloud TROPOMI/MOPITT ocean analysis to Supplement Materials. We merged parts of Fig. 10 and 11 into a new Fig. 10 which illustrates our comparison of TROPOMI/MOPITT total CO columns over oceans.

(*) Please note that latexdiff, the tool that produces the marked-up file, has issues dealing with changes in LaTeX syntax associated with citations (not dealing with changes in the citations themselves). Latexdiff can still produce a marked-up file in these cases, if executed with the option to ignore citations (--disable-citation-markup). The resulting marked-up file, as a consequence, shows "(?)" where the citations should be; we apologize for this and we would like to emphasize that most citations have remained unchanged in the manuscript. Unfortunately, we find that other references (to equations, tables, and figures) as well as tables themselves are affected in a similar manner, despite the fact that they haven't been modified (except for Fig. 10 and new Tables 4 and 5).

Responses to Review #1

We appreciate your comments. Please find our responses below. Line numbers refer to the manuscript as submitted for the discussion phase.
The authors provided a validation for the TROPOMI CO observations by using MOPITT and Atom CO measurements. I found the paper is well written. The authors demonstrated good agreement between TROPOMI and MOPITT CO observations, which is helpful for people who are interested in the sources and variations of CO from global to regional scales. I recommend the paper for publication after consideration of the minor points below.

Specific Comments:
1. Lines 154-160: It is difficult to follow this paragraph. I checked Section 2 again but didn't find the details to support the direct comparison without transformation. It would be better to provide more details here.

Thank you for this comment. New text and two tables with results from an additional analysis have been included in the manuscript to 1) better justify the direct comparisons without transformation and 2) investigate the effect on biases of the differences between MOPITT *a priori* CO profiles and TROPOMI reference CO profiles. New Section 3.1 discusses in more detail the differences between the MOPITT and TROPOMI CO retrieval algorithms, as well as the challenges these differences impose when comparing the two datasets. New Section 3.3.1 discusses the main sources of error in satellite CO retrievals; it also discusses sources of error when comparing satellite datasets, e.g., differences in *a priori* information used by each dataset and differences in vertical sensitivity (represented by the averaging kernels, or AKs) between instruments.
Determining whether or not observed differences in retrievals from these two instruments are consistent with differences in their *a priori*, AKs, and instrument noise would require knowledge of the true atmosphere during observation; this information is often unavailable, here included. Our main goal in comparing MOPITT and TROPOMI total CO column retrievals is to quantify differences between the two retrieval products available to users, rather than quantify the actual bias of either product. This goal is addressed by direct "end to end" comparisons of the two untransformed products in various regions of interest, after colocation of the MOPITT and TROPOMI retrievals. These comparisons quantify the MOPITT/TROPOMI difference statistics due to all effects: AK differences, *a priori* differences, and instrument noise.
Additionally, we now investigate the effects of differences between the *a priori*/reference information used by MOPITT and TROPOMI in their retrievals; we do so by applying a null-space adjustment (based on the MOPITT *a priori*) to TROPOMI. We present results from this additional analysis in Sections 4.1.4 and 4.2.3 and show that differences in *a priori*/reference CO profiles affect MOPITT/TROPOMI relative biases by 1-2 percentage points, well below TROPOMI's required 15% accuracy.

2. Section 4.1. The comparison between TROPOMI and MOPITT is very interesting. I have some suggestions, which may be considered in this work or the following study: 1) The differences between two datasets show obvious seasonal variabilities. What are the possible explanations? We thank you for this observation. The following text has been added to address this point (line 363): "There appears to be a seasonal component in MOPITT/TROPOMI bias values in the two hemispheric ROIs and Australia. Polluted ROIs (USA, Europe, India, China) and the Sahara do not seem to be affected (Fig. 3, 4, and 5). Biases between MOPITT and null-space adjusted TROPOMI retrievals show the same seasonal component, indicating that it is not caused by the MOPITT *a priori*. The seasonal variability of MOPITT has been validated in the past using ground-based measurements. In their comparison to NDACC data (Network for the Detection of Atmospheric Composition Change; De Maziere et al., 2018), Buchholz et al. (2017) found no significant seasonally dependent bias for MOPITT products. Hedelius et al. (2019) compared MOPITT to the TCCON dataset, reporting no persistent seasonal trend globally and some seasonal variability for individual sites. Further work will be needed to identify the origin of a possible seasonal component in MOPITT-TROPOMI bias values." 2) We know that MOPITT show some latitude-depended differences relative to surface and aircraft measurements. Do the differences between TROPOMI and MOPITT have similar latitudinal dependence? It has, indeed, been shown that V7 MOPITT TIR products exhibited a latitudinal dependence in partial CO column biases; the latitudinal dependence in total column biases was less prominent (see Fig. 2 from Deeter et al., 2019, shown below). This latitudinal dependence of biases could have been caused by issues in modeled water vapor absorption in the MOPITT TIR passband (Edwards et al., 1999) or accuracy of water vapor data used in the MOPITT retrieval (Pan et al., 1995; Wang et al., 1999). According to Deeter et al. (2019), "MOPITT V8 biases […] do not exhibit a clear latitudinal dependence"; this is particularly the case for total column values (see Fig. 6 from Deeter et al., 2019, shown below). Enhancements in the V8 retrieval algorithm addressing this issue include updated spectroscopic information used by the radiative transfer model and improved radiance bias correction. We have added wording to the MOPITT description section to clarify this point (page 5, line 116): "Here we use daytime archive MOPITT data from version 8 (Deeter et al., 2019); among other improvements, V8 products do not exhibit a latitudinal dependence in partial CO column biases observed in V7."

References
Deeter, M. N., Edwards, D. P., Francis, G. L., Gille, J. C., Mao, D., Martinez-Alonso, S., Worden, H. M., Ziskin, D., and Andreae, M. O. (2019) Radiance-based retrieval bias mitigation for the MOPITT instrument: the version 8 product, Atmospheric Measurement Techniques, 12, 4561–4580, https://doi.org/10.5194/amt-12-4561-2019.
Edwards, D. P., Halvorson, C. M., and Gille, J. C. (1999) Radiative transfer modeling for the EOS Terra satellite Measurements of Pollution in the Troposphere (MOPITT) instrument, J. Geophys. Res., 104, 16755–16775, https://doi.org/10.1029/1999JD900167.
Pan, L., Edwards, D. P., Gille, J. C., Smith, M. W., and Drummond, J. R. (1995) Satellite remote sensing of tropospheric CO and CH 4 : forward model studies of the MOPITT instrument, Appl. Opt., 34, 6976–6988, https://doi.org/10.1364/AO.34.006976.
Wang, J., Gille, J. C., Bailey, P. L., Drummond, J. R., and Pan, L. (1999) Instrument sensitivity and error analysis for the remote sensing of tropospheric carbon monoxide by MOPITT, J. Atmos. Ocean. Tech., 16, 465–474, https://doi.org/10.1175/1520-0426(1999)016%3C0465:ISAEAF%3E2.0.CO;2.

[Figure]

**Figure 2.** Latitude dependence of V7 TIR-only biases based on the HIPPO CO profiles. Results from each of the five stages of HIPPO are color-coded, as indicated by the key in the top-left panel. Large black diamonds and error bars in each panel indicate bias statistics (mean and standard deviation) representing each 30° wide latitudinal zone.

[Figure]

**Figure 6.** Latitude dependence of V8 TIR-only biases (expressed in percent) based on the HIPPO CO profiles. See caption to Fig. 2.

Thank you for your comments. Please find our responses below. Page and line numbers refer to the manuscript as submitted for the discussion phase.
This manuscript compares TROPOMI carbon monoxide retrievals to data from the MOPITT satellite and in situ airborne profiles (ATom-4). The manuscript is well written and falls into the scope of AMT. I recommend publication after the following comments have been addressed.

General Comments
My main concern is that the significantly different vertical sensitivities of the instruments and the different apriori profiles used in the algorithms are not taken into account in the comparison of the TROPOMI and MOPITT data. It is alleged that the corresponding comparison methodology is not applicable to profile scaling retrievals. However, I do not agree with this view as TROPOMI's averaging kernels (AKs) take into account that it is a profile scaling retrieval. The AK value of the i-th layer quantifies as usual the sensitivity of the total column to a change of CO in the i-th layer. It is also not a question of constraining the results with the apriori or not. If the AKs are not a direct output of the retrieval, you can simply compute them for every kind of algorithm by confronting the retrieval with simulated measurements and doing the following for each layer i: 1) change the abundance in the i-th layer, 2) perform the retrieval, 3) compare the retrieved column to the "true" column. For a meaningful comparison, at least the individual apriori profiles of both retrievals should be replaced by a common prior by using the AKs (see e.g. Section 4 of Rodgers and Connor (2003) or Appendix A of Wunch et al. (2011)). The common prior can be the TROPOMI prior, the MOPITT prior, or a different third prior. Please improve the comparison method by taking these aspects into account or give a justification why the consideration of the AKs is negligible in this analysis and prove by example that the figures of merit like the global bias between the two data sets do not critically depend on whether the individual apriori profiles are replaced by a common prior.

Thank you for this comment. New text and two tables with results from an additional analysis have been included in the manuscript to 1) better justify the direct comparisons without transformation and 2) investigate the effect on biases of the differences between MOPITT *a priori* CO profiles and TROPOMI reference CO profiles. New Section 3.1 discusses in more detail the differences between the MOPITT and TROPOMI CO retrieval algorithms, as well as the challenges these differences impose when comparing the two datasets. New Section 3.3.1 discusses the main sources of error in satellite CO retrievals; it also discusses sources of error when comparing satellite datasets, e.g., differences in *a priori* information used by each dataset and differences in vertical sensitivity (represented by the averaging kernels, or AKs) between instruments.
Determining whether or not observed differences in retrievals from these two instruments are consistent with differences in their *a priori*, AKs, and instrument noise would require knowledge of the true atmosphere during observation; this information is often unavailable, here included. Our main goal in comparing MOPITT and TROPOMI total CO column retrievals is to quantify differences between the two retrieval products available to users, rather than quantify the actual bias of either product. This goal is addressed by direct "end to end" comparisons of the two untransformed products in various

regions of interest, after colocation of the MOPITT and TROPOMI retrievals. These comparisons quantify the MOPITT/TROPOMI difference statistics due to all effects: AK differences, *a priori* differences, and instrument noise.

Additionally, we now investigate the effects of differences between the *a priori*/reference information used by MOPITT and TROPOMI in their retrievals; we do so by applying a null-space adjustment (based on the MOPITT *a priori*) to TROPOMI. We present results from this additional analysis in Sections 4.1.4 and 4.2.3 and show that differences in *a priori*/reference CO profiles affect MOPITT/TROPOMI relative biases by 1-2 percentage points, well below TROPOMI's required 15% accuracy.

Specific Comments

Page 1, Lines 4-5: TANSO-FTS-2 on the GOSAT-2 satellite (launched in 2018) is also deriving CO from solar reflected radiances in the 2.3 μm spectral region. Thank you for bringing this point to our attention; similarly, SCIAMACHY should also have been included in the list of satellite instruments that derived CO from solar reflected radiances. We have reworded the sentence as follows: "MOPITT and TROPOMI are two of only a few satellite instruments to ever derive CO from solar reflected radiances." We have also added an introduction to SCIAMACHY and TANSO-FTS-2 later on in the manuscript, please see below.

Page 1, Lines 16-17: see general comments. Please note that, for focus, we have moved the MOPITT/TROPOMI above and below cloud comparison to the Supplement Materials. Because of this, the Abstract now does not refer to this particular type of comparison. Please see response to General Comments for more details.

Page 3, Lines 49-50: TANSO-FTS-2 on the GOSAT-2 satellite is also deriving CO from solar reflected radiances in the 2.3 μm spectral region. To address this point we have reworded the sentence in lines 49-50 and introduced both SCIAMACHY and TANSO-FTS-2 as follows: "TROPOMI was, until recently, the only other operative satellite instrument retrieving CO from NIR measurements. (ENVISAT SCIAMACHY (2002-2012; Bovensmann et al., 1999) and GOSAT-2 TANSO-FTS-2 (since 2019; NIES, 2019) are two other instances.) Thus, understanding how MOPITT and TROPOMI retrievals compare to each other is important."

Page 7, Lines 157-160: see general comments. The content of these lines regarding MOPITT and TROPOMI algorithm differences has been expanded and clarified in new Section 3.1. Please see our response to General Comments for additional details.

Page 8, Line 195: The units in Eq. (1) do not match.
For added clarity and to show explicitly that units do match, we now provide the units of the constant $2.12*10^{13}$, which are molec. cm$^{-2}$ hPa$^{-1}$ ppb$^{-1}$. (Please note that Eq. 1 is now Eq. 3.)

Page 9, Lines 223-225: Why not use the actual TROPOMI averaging kernels here instead of a binary step function? Please note that, for focus, the analysis described in Section 3.2.2 has been moved to Supplement Materials. The purpose of the method described in Section 3.2.2. is to calculate the worst-case scenario errors (the maximum errors) that could be introduced by the use of modeled CO in TROPOMI retrievals over water. In a way, the method can be understood better by thinking of a step function, since it is assumed that TROPOMI sensitivity to CO above cloud top would be 1 (i.e., no modeled CO involved), while below cloud top would be 0 (i.e., only modeled CO involved). As stated

in the manuscript, this method would be most accurate in case of optically thick clouds. To explain the motivation for this section better and thus clarify this point, we have reworded the text as follows (page 9, lines 223-227): "The goal of this analysis was to calculate the maximum error caused by the use of reference CO profiles in TROPOMI retrievals over water. To this effect, we assumed that TROPOMI retrievals are only sensitive to CO above cloud top, while CO below cloud top is fully approximated by TROPOMI's scaled reference profiles. This scenario would be most accurate in case of optically thick clouds. To quantify this error, we compared TROPOMI retrievals over bodies of water (total columns and their above cloud partial column components) to their colocated MOPITT TIR counterparts."

Page 10, Lines 242-243: Why not use the actual TROPOMI averaging kernels? Please see response to previous comment. To explain the motivation for this section better and thus clarify this point, we have reworded the text as follows (Supplement Materials page 2, lines 43-44): "The goal of this analysis was to calculate the maximum error caused by the use of reference CO profiles in TROPOMI retrievals over water. To this effect, [...]"

Page 13, Lines 342-343: The negative bias could simply be a consequence of the different sensitivities and apriori profiles used in the estimation of the true atmospheric state for the two individual instruments. Thus, consideration of the averaging kernels is important. Please note for example the change in sign for the biases in Figure 6 due to the AKs. Thank you for this comment. New Table 4 shows that land MOPITT/TROPOMI bias values after accounting for *a priori* differences are very similar to (and retain the same negative sign as) the original bias values shown in Table 1 and discussed in lines 342-343. As discussed in the response to General Comments, the effect of *a priori*/reference CO profile differences on relative biases is very small, only 1-2 percentage points.

Page 14, Lines 378-380: How do these error estimates change when considering the averaging kernels and apriori profiles? Please note that, for focus, we have moved the MOPITT/TROPOMI above and below cloud comparison to the Supplement Materials. Because of this, the Discussion section now does not refer to this particular type of comparison. The manuscript does, however, still include a comparison of MOPITT/TROPOMI total CO column values over ocean plus its MOPITT/null-space adjusted TROPOMI comparison counterpart. New Table 5 shows that ocean MOPITT/TROPOMI total column bias values after accounting for *a priori* differences are very similar to the original bias values (shown in Table 3) and retain the same sign.

Page 14, Lines 384-386: In addition to the global accuracy and precision, it would also be interesting to quantify the regional relative accuracy quantifying region-to-region biases, e.g. the standard deviation of the individual biases for the regions of Figures 2-5. We assume that "region-to-region" (here and in other comments below) means "for each ROI". Please note that the bias (accuracy) and standard deviation of bias (precision) had been quantified for each ROI, both in percentage and in CO column values; results are summarized in Table 1, described in the Results section, and revisited in the Discussion section. (We assume the comment refers to Fig. 3-5.) The precision requirement of 10 % is not satisfied. Thank you for pointing this out. Calculated precision versus required precision had been discussed elsewhere in the manuscript (e.g., page 13 lines 341-342 and 366-367). However, the wording in page 14 lines 384-386 is indeed insufficient. For clarity, the sentence has been reworded to: "Our results show that the accuracy of TROPOMI retrievals with respect to MOPITT and ATom far exceeds Sentinel-5P mission requirements (Veefkind et al., 2012; Landgraf et al., 2016). The precision values calculated for some of the ROIs analyzed surpass the target value by a few percent."

Page 14, Line 390: To validate TROPOMI retrievals over land, ground-based measurements from the Total Carbon Column Observing Network (TCCON), which are calibrated using aircraft profiles, can also be used. In contrast to aircraft data this would also allow the validation of seasonal variability at a fixed location. We agree, thanks for bringing this point up. To address this comment we have reworded/added the following text in the manuscript: "To that end, *in situ* data from other airborne measurement programs are required. Ground-based measurements (e.g., NDACC, TCCON) could also be used; this would allow the validation of seasonal variability at fixed locations."

Page 14, Line 392-393: I would call a comparison to other satellite data sets verification instead of validation. Thank you for this comment. We have reworded this sentence as follows: "The MOPITT dataset represents the longest global CO record available (2000-present); because of extensive validation efforts with respect to *in situ* measurements and comparisons with other satellite datasets, it is well characterized." Has the seasonal variability of MOPITT also been validated, e.g. by using TCCON or NDACC ground-based measurements? MOPITT has been compared in the past to ground measurements, as discussed in page 3 line 52. (In that line, for simplicity and focus, neither the names of these ground networks nor the names of the individual satellite datasets used in previous MOPITT validations are provided.) We have added elsewhere (page 13, line 363) the following text describing these previous efforts: "The seasonal variability of MOPITT has been validated in the past using ground-based measurements. In their comparison to NDACC data (Network for the Detection of Atmospheric Composition Change; De Maziere et al., 2018), Buchholz et al. (2017) found no significant seasonally dependent bias for MOPITT products. Hedelius et al. (2019) compared MOPITT to the TCCON dataset, reporting no persistent seasonal trend globally and some seasonal variability for individual sites." Relevant references have been added to the manuscript.

Page 14, Line 395-396: TANSO-FTS-2 on the GOSAT-2 satellite is also deriving CO from solar reflected radiances in the 2.3 μm spectral region. Thanks for pointing that out. We have reworded the sentence as follows: "Furthermore, TROPOMI and MOPITT were, until TANSO-FTS-2 became operational in 2019, the only working satellite instruments retrieving CO from NIR solar-reflected radiances."

Page 14, Line 399: Please replace "do not fully account for" by "do not account for". Done.

Page 15, Line 408: What about transport of CO from major sources in coastal regions to the ocean? Thanks for this observation. We have address the comment as follows: "Since there are no major CO sources over water, CO values closer to the surface (and, therefore, most likely to be below cloud top) tend to be spatially homogeneous and stable through time. Thus, they are well characterized by the reference profiles. (Caution should be exercise in case of sporadic CO sources near open water, e.g., fires near a coastline, which could in some cases result in plumes transported off the coast and below cloud top. Larger errors could occur in such retrievals over water, if sources were not well represented in the TM5 model.) "

Page 15, Line 411-412: or validation with ground-based measurements (TCCON or NDACC). Reworded to "These errors require further characterization with colocated *in situ* data and ground measurements over land."

Figures 3-5: Please add regional mean bias and standard deviation of the differences to all individual subplots. Figures 3-5 currently show daily mean of regional bias for each ROI. Please note that regional mean bias values for each ROI and each MOPITT product (TIR, NIR, and TIR+NIR) are shown in Fig.

13. Adding this information to Fig. 3-5 would be redundant and would make these figures harder to read. Regional standard deviations for each ROI and each MOPITT product are also shown in Fig. 13; adding this information to Fig. 3-5 would make these figures harder to read.

Figures 13-14: Please add the standard deviation of the individual regional biases as a measure of the region-to-region bias to the plots (for TIR, NIR, and TIR+NIR). Please note that Fig. 13 shows results for colocated retrievals while Fig. 14 shows results for non colocated retrievals. The solid black lines in Fig. 13 represent, for each ROI, the standard deviation derived from individual biases between each pair of colocated observations. We have added a few words early in the manuscript (Methods section, page 7, line 168) to clarify this point: "We quantified, among others, daily bias (i.e., accuracy) and standard deviation (i.e., precision; calculated from individual biases between each pair of colocated observations) between TROPOMI and each of the three MOPITT products (TIR, NIR, and TIR+NIR)." In contrast, the dashed lines (not "solid lines"; caption has been corrected accordingly, here and in Supplement Materials) in Fig. 14 represent, for each ROI, ±1 standard deviation of mean daily relative biases (i.e., inter-daily bias variability). In this case a standard deviation cannot be calculated from individual biases between each pair of colocated observations, since no colocation was performed.

Table 1: Please add region-to-region biases in case "all ROIs". Please note that Table 1 already contains, for each ROI, the biases as well as the standard deviation of the individual biases, both in percentage and in CO column values.

Technical Corrections
Page 3, Line 50: Please rephrase "is key." Reworded to "is important".
Page 8, Line 195: Please replace $\sum_{i=1}^{i=n} ...$ by $\sum_{i=1}^{n} ...$ in Eq.(1). Thank you. The equation (now Eq. 3) has been slightly simplified and now it does not include a summation symbol.
Page 13, Line 340: Please add the unit % for the relative biases. Done.

Thank you for this comment. New text and two tables with results from an additional analysis have been included in the manuscript to 1) better justify the direct comparisons without transformation and 2) investigate the effect on biases of the differences between MOPITT *a priori* CO profiles and TROPOMI reference CO profiles. New Section 3.1 discusses in more detail the differences between the MOPITT and TROPOMI CO retrieval algorithms, as well as the challenges these differences impose when comparing the two datasets. New Section 3.3.1 discusses the main sources of error in satellite CO retrievals; it also discusses sources of error when comparing satellite datasets, e.g., differences in *a priori* information used by each dataset and differences in vertical sensitivity (represented by the averaging kernels, or AKs) between instruments.

Determining whether or not observed differences in retrievals from these two instruments are consistent with differences in their *a priori*, AKs, and instrument noise would require knowledge of the true atmosphere during observation; this information is often unavailable, here included. Our main goal in comparing MOPITT and TROPOMI total CO column retrievals is to quantify differences between the two retrieval products available to users, rather than quantify the actual bias of either product. This goal is addressed by direct "end to end" comparisons of the two untransformed products in various regions of interest, after colocation of the MOPITT and TROPOMI retrievals. These comparisons quantify the MOPITT/TROPOMI difference statistics due to all effects: AK differences, *a priori* differences, and instrument noise.

Additionally, we now investigate the effects of differences between the *a priori*/reference information used by MOPITT and TROPOMI in their retrievals; we do so by applying a null-space adjustment (based on the MOPITT *a priori*) to TROPOMI. We present results from this additional analysis in

Sections 4.1.4 and 4.2.3 and show that differences in *a priori*/reference CO profiles affect MOPITT/TROPOMI relative biases by 1-2 percentage points, well below TROPOMI's required 15% accuracy.

2. Lines 162-163: What is the impact of the differences in the overpass times of TROPOMI and MOPITT when selecting "collocated" pairs of data? Quantifying this for the ROIs selected in the study would be helpful for interpreting the results of the intercomparison. Quantifying the effect of differences in passing times in CO retrievals is an interesting topic, but it is outside the scope of this work. Please note that validation papers allow time differences substantially larger than the 3 hours between MOPITT and TROPOMI, e.g., Deeter et al., 2019 (12 hours); Clerbaux et al, 2008 (24 hours). The lifetime of CO (several weeks) is much greater than the time difference between MOPITT and TROPOMI passing times. Differences in total CO column amounts due to transportation would be equally likely to be positive or negative; thus, they would not contribute to an apparent bias between the two products.

3. Lines 307-308: What is the implication of the tendency of the reference profiles to have too much CO near the surface for the intercomparison with MOPITT, considering that no attempt is made to mitigate potential biases arising from the a priori? Thank you for this comment. Please see response to Comment #1 above. As explained there, the manuscript now includes text describing (and results from) an additional analysis where we quantify the effect on biases of the differences between MOPITT *a priori* CO profiles and TROPOMI reference CO profiles. We show that differences in *a priori*/reference CO profiles affect MOPITT/TROPOMI relative biases by 1-2 percentage points, well below TROPOMI's required 15% accuracy.

4. Lines 354-357: It is certainly possible that the differences in overpass times could contribute to these biases over Africa, but this can be confirmed with a model, for example. Modeling the effect of differences in passing times in CO retrievals is an interesting topic, but it is outside the scope of this work. Also, please note that lines 354-357 discuss results obtained for the China ROI; furthermore, please note that the ROIs analyzed in this work do not include fire regions in Africa. Furthermore, what about the impact of the different vertical sensitivities of the measurements here? It seems critical to me to account for the influence of the averaging kernels before speculating that these differences could be due to temporal variations in the African fires. Thank you for this comment. Please see response to Comment #1 for more details regarding additional text now included in the manuscript to address this point.

Technical Comments

1) Line 46: This is not the first use of the acronym MOPITT. Thank you for catching this. We have reworded lines 33-35 to include definitions of TROPOMI, MOPITT, and ATom the first time they are mentioned in the Introduction: "The aim of this work is to facilitate the extension of the current satellite record with newly available TROPOMI (TROPOspheric Monitoring Instrument) measurements by evaluating those with respect to satellite MOPITT (Measurements Of Pollution In The Troposphere) and *in situ* ATom (Atmospheric Tomography mission) CO data." Also, the MOPITT and ATom acronym definitions in lines 46-47 have been removed and the sentence reworded to: "Here we analyze daily global TROPOMI retrievals acquired between 7 November 2017 and 10 March 2019 with respect to MOPITT and ATom."

2) Line 110: Please insert "the" before "total column AK". Please note line 110 does not contain the text "total column AK". That text appears, though, in lines 109 and 111; we have added "the" to the

latter occurrence. That sentence now reads: "With respect to vertical sensitivity, the total column AK for the NIR-only product are most similar in shape to the TROPOMI total column AK"

3) Line 119: Please make it clear that "(∼480; note 1 km resolution)" here is referring to the number of MODIS observation, and that these observations have a resolution of 1 km. Thank you for this comment. For increased clarity that sentence has been reworded to: "
[revised manuscript text omitted]

**S1    TROPOMI versus MOPITT over land: non-colocated retrievals**

Here we describe results from the analysis of daily (from 7 November 2017 to 10 March 2019) non-colocated TROPOMI and MOPITT retrievals over 8 ROIs  (regions of interest). Polluted ROIs include: south-eastern USA (USA; 35°N, 95°W to 40°N, 75°W), central Europe (Europe; 45°N, 0°E to 55°N, 15°E), northern half of the Indian Subcontinent (India; 20°N, 70°E to 30°N, 95°E), and north-eastern China (China; 30°N, 110°E to 40°N, 123°E). Clean ROIs are: northern Africa and Arabia (Sahara; 15°N, 20°W to 30°N, 50°E) and western Australia (Australia; 32°S, 112°E to 17°S, 138°E). Two additional ROIs represent most of the northern and southern (N and S) hemispheres (0°N to 60°N and 60°S to 0°N, respectively). TROPOMI and MOPITT retrievals were filtered to keep only clear daytime data over land. Daily mean retrievals for each dataset as well as relative bias between TROPOMI and each of the three MOPITT products (TIR, NIR, and TIR+NIR) were calculated; relative bias = 100*(TROPOMI-MOPITT)/MOPITT. By utilizing non-colocated retrievals we maximized the size and diversity of the populations analyzed. Results from this analysis are summarized in Fig. S1.

**S1.1    TROPOMI versus MOPITT TIR**

Results summarized in Fig. S2 show that during the ~1.5 year analyzed, TROPOMI and MOPITT TIR total CO column retrievals were close to each other both in magnitude and temporal variation. The two datasets show strong differences between clean ROIs (Sahara and Australia; 10-20 x $10^{17}$ molec. cm$^{-2}$) and highly polluted ROIs (India and China; up-to-40 x $10^{17}$

molec. cm$^{-2}$). They also show the expected differences between the two hemispheres: retrievals are, overall, lower in the S Hemisphere (10-20 x 10$^{17}$ molec. cm$^{-2}$ versus 16-22 x 10$^{17}$ molec. cm$^{-2}$) due to less land area, population, and industrial activity. Both TROPOMI and MOPITT TIR show equivalent seasonal variability. ROIs located in the Northern hemisphere present an absolute maximum during boreal winter and a relative maximum in late boreal summer. The absolute maximum is consistent with winter CO accumulation due to shorter days and larger zenithal angles, resulting in less photolysis, and to increased emissions due to biomass burning north of the Equator in Africa. The relative maximum is most likely due to fire emissions. Conversely, seasonal trends in Southern hemisphere ROIs show a maximum in September-October, consistent with CO accumulation during austral winter and emissions from biomass burning S of the equator. Daily relative bias values are generally within a ±10 % range for all the ROIs except the two most polluted (India and China), where most values are between -20 to +40 %. When averaged over time (Table S1), relative biases are between -10.07 % (S Hemisphere) and 11.73 % (China), with a mean for all the ROIs of -3.81 %. We note that biases for most ROIs are predominantly negative, except for China.

**S1.2 TROPOMI versus MOPITT NIR**

Figure S3 shows daily results from the comparison of non-colocated TROPOMI and MOPITT NIR land retrievals; time-averaged results are summarized in Table S1. The ranges of daily mean retrievals and seasonal trends observed in each ROI are in general analogous to those described in Sect. S1.1. Relative bias values averaged for the period analyzed range between -10.60 % (S Hemisphere) and 6.88 % (China), while the mean for all the ROIs is -2.99 %. Bias values for the Sahara ROI (mostly in the -5 to 10 % range) contrast strongly with those shown in Fig. S2.g (-10 to -5 %). For all the other ROIs, relative biases with respect to MOPITT NIR are broadly similar in magnitude to those respect MOPITT TIR, albeit the former present larger oscillations along time. This is consistent with the MOPITT NIR retrievals being more sensitive to geophysical noise due to changes in albedo during MOPITT observation associated with spacecraft motion (Deeter et al., 2011).

**S1.3 TROPOMI versus MOPITT TIR+NIR**

Daily results from non-colocated TROPOMI and MOPITT TIR+NIR retrievals are shown in Fig. S4; temporally averaged results are summarized in Table S1. Results are similar to those described in Sect. S1.1 in terms of daily means, seasonal trends, and relative biases. The latter range between -9.96 % (S Hemisphere) and 12.73 % (China); the mean for all ROIs is -3.50 %.

**S2 TROPOMI versus MOPITT TIR over water: above/below cloud analysis**

The goal of this analysis was to calculate the maximum error caused by the use of reference CO profiles in TROPOMI retrievals over water. To this effect, we assumed that TROPOMI retrievals are only sensitive to CO above cloud top, while CO below cloud top is fully approximated by TROPOMI's scaled reference profiles. This scenario would be most accurate in case of optically thick clouds. To quantify this error, we compared TROPOMI retrievals over bodies of water (total columns and their above cloud partial column components) to their colocated MOPITT TIR counterparts. For each TROPOMI observation, a

partial above cloud column was calculated by subtracting from the reported total TROPOMI column the below cloud partial column of its colocated, scaled TROPOMI reference profile, available in a 25-level vertical grid. Scaling factors produced in the TROPOMI retrieval process are not included in the TROPOMI product; we obtained those by dividing each reported TROPOMI total CO column retrieval by the total CO column of its colocated reference profile. Total and partial above cloud column values were also calculated for the colocated MOPITT TIR profiles interpolated to match the 25-level vertical grid of the reference profiles.

Figure S5 and Table S2 summarize results from our comparison of colocated TROPOMI and MOPITT TIR retrievals over bodies of water in the N Hemisphere ROI. The top panels in Fig. S5 illustrate a comparison between total column (above and below cloud top) and partial column (above cloud top) retrievals for a single day, 1 January 2018. Partial column values from TROPOMI and MOPITT are more strongly correlated in this particular date, as shown by a larger R (0.87 versus 0.73) and a smaller relative bias (2.77 versus 2.92 %). The bottom panels in Fig. S5 summarize similar daily results for the entire ~1.5-year period analyzed. Relative biases between TROPOMI and MOPITT TIR for total or partial columns are small (in the -2 to 11 % range, ~4 % on average) and follow the same temporal patterns; their differences (total column bias - partial column bias) range from -1.79 to 1.56 p.p. (percentage points), with a -0.53 p.p. mean. Standard deviation values are on average around 13-15 %.

Similar results for the S Hemisphere ROI are summarized in Fig. S6 and Table S2. Partial column values for 1 January 2018 (Fig. S6.b) have a larger R (0.84 versus 0.79) and appear more strongly correlated than their total column counterparts (Fig. S6.a). They, however, show a larger relative bias (2.16 versus 0.36 %). Similar results for the entire period analyzed (Fig. S6.c and .d) indicate that relative biases for either total or partial columns are similarly small, ranging from -5 to 7 (~3 % mean). Their differences are in the -3.62 to 0.97 p.p. range, with a -1.02 p.p. mean. Standard deviations are in the 18-21 % range.

Based on the difference in relative bias between the total (above and below cloud) and partial (above cloud) column analyses, we estimate that approximating TROPOMI CO below cloud top by scaled reference profiles results, on average, in a ~-0.78 p.p. error. This approach would be most accurate in the presence of optically thick clouds which would preclude TROPOMI sensitivity below cloud top.

**S3   TROPOMI versus ATom-4 over water: above/below cloud analysis**

Results from an analysis of colocated TROPOMI and true (unsmoothed) ATom-4 profiles over bodies of water performed for the period between 24 April and 21 May 2018 are summarized in Fig. S7 and Table S3. Colocation criteria were same day acquisition and horizontal distance $\leq$ 50 km; each ATom-4 profile was paired with the closest valid TROPOMI retrieval that met the colocation criteria.

 The goal of this analysis was to calculate the maximum error caused by the use of reference CO profiles in TROPOMI retrievals over water. To this effect, we assumed that TROPOMI retrievals are only sensitive to CO above cloud top, while CO below cloud top is fully approximated by TROPOMI's scaled model-based reference profiles. This scenario would be most accurate in case of optically thick clouds. To quantify

this error, we compared TROPOMI retrievals over bodies of water (total columns and their above cloud partial column components) to their colocated ATom-4 counterparts. Complete (e.g., from the surface to the top of the atmosphere) ATom-4 CO profiles were generated following the standard method for MOPITT validation with airborne data, as described in the main article. The complete profiles were then interpolated to match the TROPOMI reference profile 25-level vertical grid. ATom total CO column values were calculated applying Eq. 3 in main article. The corresponding ATom partial column values were also calculated, including only the layers above cloud top. For each TROPOMI observation, a partial above cloud column was calculated by subtracting from the reported total TROPOMI column the below cloud partial column of its colocated, scaled TROPOMI reference profile.

Fig. S7.a shows total CO column retrievals which, for TROPOMI, according to our assumption, would include a measured component (partial column above cloud top) and a reference component (partial column below cloud top). TROPOMI and ATom-4 total CO column values show very strong correlation (R = 0.93, slope of linear fit = 0.96) and a small negative relative bias (-4.76 %) indicative of slightly low TROPOMI values with respect to ATom-4. Figure S7.b shows results for partial (above cloud) CO column values. The relative bias in this case is closer to zero (-1.11 %) and the linear fit has a slightly larger R (0.95), indicative of an even stronger correlation between the above-cloud-only component of the two datasets; the slope of the linear fit is slightly lower (0.92). We interpret the difference between these two relative bias values (-3.65 p.p.) as an estimate of the error introduced by assuming that below-cloud partial CO columns can be approximated by TROPOMI scaled CO reference profiles. Results from this analysis characterize a worst-case scenario (where TROPOMI has no sensitivity to CO below cloud top) and they complement results from the TROPOMI versus ATom-4 analysis presented in the main article, where it is assumed that TROPOMI has some sensitivity to CO below cloud top.

**100 References**

Deeter, M. N., Worden, H. M., Gille, J. C., Edwards, D. P., Mao, D., and Drummond, J. R.: MOPITT multispectral CO retrievals: Origins and effects of geophysical radiance errors, JOURNAL OF GEOPHYSICAL RESEARCH-ATMOSPHERES, 116, https://doi.org/10.1029/2011JD015703, 2011.

[Figure]

**Figure S1.** Summary of non-colocated land comparison results. Colored bars represent relative bias between TROPOMI and each of the three MOPITT products (TIR, NIR, and TIR+NIR).  Dashed lines show ± 1 standard deviation of mean daily relative biases (i.e., inter-daily bias variability). Same as Fig. 13 in the main article.

[Figure]

**Figure S2.** Comparison of non-colocated land retrievals from TROPOMI (pink) and MOPITT TIR (green) for each ROI analyzed. Filled circles show daily mean. Thin purple lines indicate daily relative bias between the two datasets, thick purple lines are a 11-day smoothed version with high-frequency variability removed. Gray bars show periods without MOPITT measurements because of hot calibrations (March and October 2018) or a safe mode maneuver (October-November 2018). Note that for the India and China ROIs the relative bias scale is different than for the other ROIs.

[Figure]

**Figure S3.** Comparison of non-colocated land retrievals from TROPOMI (pink) and MOPITT NIR (blue) for each ROI analyzed. See caption to Fig. S2 for details.

[Figure]

**Figure S4.** Comparison of non-colocated land retrievals from TROPOMI (pink) and MOPITT TIR+NIR (black) for each ROI analyzed. See caption to Fig. S2 for details.

[Figure]

**Figure S5.** Comparison of colocated retrievals over bodies of water from TROPOMI and MOPITT TIR for the N Hemisphere ROI. a) Total CO column values (above and below cloud top) for a single day, 1 January 2018. b) Partial CO column values (above-cloud only) for the same day. c) Compilation of means and relative biases of total CO column values (above and below cloud top) from 7 November 2017 to 10 March 2019. d) Same for partial CO column values (above-cloud only).

[Figure]

**Figure S6.** Comparison of colocated retrievals over bodies of water from TROPOMI and MOPITT TIR for the S Hemisphere ROI. a) Total CO column values (above and below cloud top) for a single day, 1 January 2018. b) Partial CO column values (above-cloud only) for the same day. c) Compilation of means and relative biases of total CO column values (above and below cloud top) from 7 November 2017 to 10 March 2019. d) Same for partial CO column values (above-cloud only).

[Figure]

**Figure S7.** Comparison of colocated retrievals over bodies of water from TROPOMI and true ATom-4 (unsmoothed), performed for the period between 24 April and 21 May 2018. a) Total column retrievals (above and below cloud top), b) Partial column retrievals (above cloud top only).

**Table S1.** Statistics from non-colocated TROPOMI (T) versus MOPITT (M) retrievals over land for the period between 7 November 2017 and 10 March 2019. Relative bias in %. Column bias in units of $10^{17}$ molec. cm$^{-2}$.

| | | TROPOMI vs MOPITT$_{TIR}$ | TROPOMI vs MOPITT$_{NIR}$ | TROPOMI vs MOPITT$_{TIR+NIR}$ |
|---|---|---|---|---|
| N Hemisphere | Relative Bias | -3.88 | 0.19 | -3.91 |
| | Column Bias | -0.74 | 0.04 | -0.75 |
| | Mean Daily Samples (T, M) | 151685, 15716 | 151685, 15855 | 151685, 14764 |
| S Hemisphere | Relative Bias | -10.07 | -10.60 | -9.96 |
| | Column Bias | -1.55 | -1.69 | -1.53 |
| | Mean Daily Samples (T, M) | 26551, 6287 | 26551, 6323 | 26551, 5992 |
| USA | Relative Bias | -4.73 | -8.77 | -3.58 |
| | Column Bias | -1.07 | -1.99 | -0.84 |
| | Mean Daily Samples (T, M) | 1559, 144 | 1559, 143 | 1564, 142 |
| Europe | Relative Bias | -4.65 | -5.78 | -4.77 |
| | Column Bias | -1.00 | -1.20 | -1.04 |
| | Mean Daily Samples (T, M) | 1680, 146 | 1680, 146 | 1680, 142 |
| India | Relative Bias | -2.91 | -1.21 | -2.20 |
| | Column Bias | -0.98 | -0.68 | -0.92 |
| | Mean Daily Samples (T, M) | 3831, 654 | 3822, 657 | 3852, 624 |
| China | Relative Bias | 11.73 | 6.88 | 12.73 |
| | Column Bias | 2.55 | 1.20 | 2.80 |
| | Mean Daily Samples (T, M) | 1395, 197 | 1392, 198 | 1395, 191 |
| Sahara | Relative Bias | -8.01 | 1.64 | -7.96 |
| | Column Bias | -1.50 | 0.27 | -1.50 |
| | Mean Daily Samples (T, M) | 50605, 4117 | 50605, 4143 | 50605, 3872 |
| Australia | Relative Bias | -7.98 | -6.26 | -8.35 |
| | Column Bias | -1.20 | -0.90 | -1.26 |
| | Mean Daily Samples (T, M) | 5918, 1311 | 5918, 1316 | 5918, 1263 |
| Mean all ROIs | Relative Bias | -3.81 | -2.99 | -3.50 |
| | Column Bias | -0.69 | -0.62 | -0.63 |
| | Mean Daily Samples (T, M) | 30403, 3572 | 30402, 3598 | 30406, 3374 |

**Table S2.** Colocated TROPOMI versus MOPITT TIR CO retrievals over bodies of water: Statistics from above/below cloud analysis performed for the period between 7 November 2017 and 10 March 2019. Total column = above and below cloud top. Partial column = above cloud top. Relative bias and standard deviation in %. Column bias and standard deviation in units of $10^{17}$ molec. cm$^{-2}$.

| | | TROPOMI vs MOPITT$_{TIR}$ Total Column | | TROPOMI vs MOPITT$_{TIR}$ Partial Column |
|---|---|---|---|---|
| N Hemisphere | Relative Bias±St. Dev. | 3.82±13.27 | | 4.35±14.72 |
| | Column Bias±St. Dev. | 0.53±2.35 | | 0.48±2.04 |
| | Mean Daily Colocated Pairs | 127360 | | 127360 |
| | Change in Relative Bias (p.p.) | | -0.53 | |
| S Hemisphere | Relative Bias±St. Dev. | 2.14±18.15 | | 3.16±21.49 |
| | Column Bias±St. Dev. | 0.19±2.38 | | 0.24±2.14 |
| | Mean Daily Colocated Pairs | 164935 | | 164935 |
| | Change in Relative Bias (p.p.) | | -1.02 | |
| Mean both Hemispheres | Relative Bias, St. Dev. | 2.98±15.71 | | 3.76±18.11 |
| | Column Bias±St. Dev. | 0.36±2.37 | | 0.36±2.09 |
| | Mean Daily Colocated Pairs | 146148 | | 146148 |
| | Change in Relative Bias (p.p.) | | -0.78 | |

**Table S3.** Comparison of colocated retrievals over bodies of water from TROPOMI and true ATom-4 (unsmoothed): Statistics from above/-below cloud analysis performed for the period between 24 April and 21 May 2018. Relative bias in %. Column bias in units of $10^{17}$ molec. cm$^{-2}$.

| | | TROPOMI vs true ATom-4 Above & Below Cloud Top | TROPOMI vs true ATom-4 Above Cloud Top |
|---|---|---|---|
| Atlantic/Pacific | Relative Bias±St. Dev. | -4.76±11.15 | -1.11±12.92 |
| | Column Bias±St. Dev. | -0.89±1.80 | -0.17±1.51 |
| | Number of Colocated Pairs | 103 | 103 |
| | Change in Relative Bias (p.p.) | -3.65 | |